# Chip collection of hepatocellular carcinoma based on O₂ heterogeneity from patient tissue

Sewoom Baek [1,2,10], Hyun-Su Ha [2,3,10], Jeong Su Park [4,10], Min Jeong Cho [5], Hye-Seon Kim [2,6], Seung Eun Yu [2], Seyong Chung [2], Chansik Kim [1,2], Jueun Kim [1,2], Ji Youn Lee [1,2], Yerin Lee [2], Hyunjae Kim [2], Yujin Nam [2], Sungwoo Cho [2], Kyubae Lee [7], Ja Kyung Yoon [8], Jin Sub Choi [9], Dai Hoon Han [9] ✉ & Hak-Joon Sung [1,2] ✉

Hepatocellular carcinoma frequently recurs after surgery, necessitating personalized clinical approaches based on tumor avatar models. However, location-dependent oxygen concentrations resulting from the dual hepatic vascular supply drive the inherent heterogeneity of the tumor microenvironment, which presents challenges in developing an avatar model. In this study, tissue samples from 12 patients with hepatocellular carcinoma are cultured directly on a chip and separated based on preference of oxygen concentration. Establishing a dual gradient system with drug perfusion perpendicular to the oxygen gradient enables the simultaneous separation of cells and evaluation of drug responsiveness. The results are further cross-validated by implanting the chips into mice at various oxygen levels using a patient-derived xenograft model. Hepatocellular carcinoma cells exposed to hypoxia exhibit invasive and recurrent characteristics that mirror clinical outcomes. This chip provides valuable insights into treatment prognosis by identifying the dominant hepatocellular carcinoma type in each patient, potentially guiding personalized therapeutic interventions.

Hepatocellular carcinoma (HCC) is the most common type of primary liver cancer and the second leading cause of cancer-related mortality worldwide, with its incidence on the rise[1,2]. Although liver resection remains the preferred treatment for achieving long-term survival, recurrence poses a challenge for more than half of HCC patients[3,4]. As a result, there is a growing interest in HCC avatar models to tailor clinical strategies specific to HCC types. Recent efforts have focused on establishing patient-derived xenografts (PDX) by implanting cancer

[1]Department of Brain Korea 21 FOUR Project for Medical Science, Yonsei University College of Medicine, 50-1 Yonsei-ro, Seodaemun-gu, Seoul 03722, Republic of Korea. [2]Department of Medical Engineering, Yonsei University College of Medicine, 50-1 Yonsei-ro, Seodaemun-gu, Seoul 03722, Republic of Korea. [3]Division of Cardiology, Severance Cardiovascular Hospital, Yonsei University College of Medicine, 50-1 Yonsei-ro, Seodaemun-gu, Seoul 03722, Republic of Korea. [4]Department of Severance Biomedical Science Institute, Yonsei University College of Medicine, 50-1 Yonsei-ro, Seodaemun-gu, Seoul 03722, Republic of Korea. [5]Department of Clinical Pharmacology & Therapeutics, Catholic University of Korea, Seoul St. Mary's Hospital, 222, BanpoDaero, Seocho-gu, Seoul 06591, Republic of Korea. [6]Department of Anesthesiology, Perioperative, and Pain Medicine, Brigham and Women's Hospital, Harvard Medical School, Boston, MA, USA. [7]Department of Biomedical Materials, Konyang University, 158, Gwanjeodong-ro, Seo-gu, Daejeon 35365, Republic of Korea. [8]Department of Radiology, Severance Hospital, Research Institute of Radiological Science, Center for Clinical Imaging Data Science, Yonsei University College of Medicine, 50-1 Yonsei-ro, Seodaemun-gu, Seoul 03722, Republic of Korea. [9]Department of Surgery, Division of Hepato-biliary and Pancreatic Surgery, Yonsei University College of Medicine, 50-1 Yonsei-ro, Seodaemun-gu, Seoul 03722, Republic of Korea. [10]These authors contributed equally: Sewoom Baek, Hyun-Su Ha, Jeong Su Park. ✉e-mail: dhhan@yuhs.ac; hj72sung@yuhs.ac

cells into mice or using cell spheroid models as organoids with patient cancer tissues[5–7]. However, these models often fail to accurately replicate the diverse oxygen gradients found within the HCC tumor microenvironment. HCC is unique in that it receives a dual blood supply, with oxygen-rich blood from the hyperoxic hepatic artery (25%, 4.4-5.2 mg L$^{-1}$) and oxygen-poor blood from the hypoxic portal vein (75%, 0.6 ~ 0.7 mg L$^{-1}$), resulting in varying oxygen levels within the liver (1.5-2.0 mg L$^{-1}$)[8,9]. These location-dependent oxygen levels contribute to the heterogeneous nature of HCC tumors[10,11] and the heterogeneity between patients[12]. However, most chips and PDX models used in HCC research expose HCC cells to either uniform normoxic 18% O$_2$ (7.76 mg L$^{-1}$) or hypoxic 0.1% O$_2$ conditions. This approach leads to lower success rates (20%) in HCC organoid and PDX studies[13,14] compared to colorectal and gynecologic cancer models (90%)[15,16]. The heterogeneity of HCCs highlights the importance of adjusting IC50 doses and types of cancer drugs administered based on the oxygen levels of the tumor environment. Targeting drug treatment to HCC types that are resistant to current therapies can improve the accuracy of prognosis for individual patients. Therefore, it is crucial to separate and collect HCC types based on their preference for oxygen levels and adjust drug doses accordingly. Additionally, a dual gradient approach that simultaneously evaluates HCC responsiveness to different oxygen levels and drug concentrations becomes necessary to fully understand the impact of these variables.

In the clinical setting, oxygen level-dependent types of HCC are identified using magnetic resonance imaging (MRI) and computed tomography (CT)[17–19]. These imaging techniques allow for the detection of variations in the arterial, portal, and delayed phases of the dynamic hepatic vascular supply. Typical HCCs exhibit contrast enhancement during the arterial phase, indicating an oxygen-rich environment fueled by an abundant supply of arterial blood from the hepatic artery. Conversely, irregular rim-enhanced (IRE) HCCs only display contrast enhancement at the tumor border, suggesting a deficiency of oxygen within the tumor. Hypoxia contributes to the aggressive nature of IRE HCCs, rendering them resistant to drug treatments and resulting in high recurrence rates with a poor prognosis[20,21]. The irregular pattern of MRI contrast also reveals the heterogeneous distribution of oxygen levels within a single HCC[22]. This heterogeneity significantly increases the rates of recurrence and mortality, despite ongoing advancements in drug development and mechanistic discovery[23]. Therefore, there is an urgent need for a translational toolbox that can analyze these heterogeneities and optimize drug treatment strategies. A promising approach involves the generation of dual gradients based on oxygen levels and drug concentrations. This approach drives cell migration in a desired direction, allowing for the investigation of drug responses within an effective range of concentrations[24–26]. In this system, molecular signatures serve as reliable trackers. Hypoxia triggers tumor cells to (i) express hypoxia-inducible factor (HIF) family proteins, which promotes invasive behavior. When HIF-1α targets carbonic anhydrase IX (CAIX), the resulting complex catalyzes the conversion of CO$_2$ to bicarbonate and protons, leading to tumor acidosis. This alteration causally enhances cell motility, invasiveness, and metastasis[27]. HIF-1α also upregulates the expression of multi-drug resistance genes (*ABCB1* and *ABCC2*), facilitating the efflux of drugs from tumor cells[28]. Under hypoxic conditions, (ii) the expression of the *GSTP1* gene also increases to reduce oxidative stress during drug treatment[29]. (iii) The activities of metalloproteinases (MMP)−1 and 9 increase, along with the expression of collagen lysis markers (*CD44* and *Vimentin*)[30], indicating extracellular matrix (ECM) remodeling. (iv) The promotion of angiogenic markers, such as vascular endothelial growth factor receptor (*VEGFR*) and *CD34*, triggers metastasis[31]. These molecular signatures suggest that hypoxia effectively isolates drug-resistant cancer cells from the heterogeneous tumor mix, contrasting with the current drug treatment paradigms where normoxia filters susceptible types. This study examines the

translational applications of the HCC avatar chip by creating a dual gradient of oxygen and drug in orthogonal directions. This is accomplished based on clinical evidence of concentration ranges. Tissue samples from both typical and IRE HCC patients are cultured directly on the chip. A gelatin hydrogel is used as the culture matrix, which controls diffusion capacity (<200 μm) through capillary-like microchannel networks[32,33]. The Soluplus® polymer is used as the sacrificial material, taking advantage of its lower critical solution temperature (LCST) property to form the channel network[34]. Computational fluid dynamics (CFD) is utilized as a predictive tool to model and calculate gradient parameters, such as media oxygen perfusion, gel permeability, microchannel porosity, and cellular oxygen consumption. The chip results are validated by implanting the same tissue-containing chips into the ischemic limb (hypoxia), subcutaneous site (sub-hypoxia), and normal limb (normoxia) of mice, thus creating a multi-spot PDX model. The chip effectively separates and collects different types of HCC while preserving the original tissue. This allows for the expression of invasiveness (CAIX)[35,36], stemness (cytokeratin 19, K19)[37], and angiogenesis (CD34)[38] markers. The consistency of results between the chip and the PDX model demonstrates that hypoxia-preferring IRE HCCs exhibit greater invasiveness, drug resistance, and recurrence compared to normoxia-preferring typical types. The chip shows promise in advancing precision medicine by customizing drug treatments for individual patients.

## Results

### HCC patient tissue

Over the past three years, HCC tissue has been collected from a total of 12 patients through surgical intervention (Supplementary Fig. 1a). These HCC samples were categorized into two groups based on MRI findings: typical ($n = 4$) and IRE ($n = 8$). The tumor mass and contrast signal intensity were analyzed using the segmentation plot (Supplementary Fig. 1b). At diagnosis, only age and tumor necrosis showed significant differences between the two groups, with the median age and tumor necrosis factors of the typical type being 53.5 years old and 2.5%, respectively, compared to 68.4 years old and 16.9% for the IRE type. Among the eight IRE patients, two (25%) experienced HCC recurrence, while none was observed in the typical group. Other clinical factors, including tumor size, viral infection, portal vein involvement, microvascular invasion, tumor markers (AFP, PIVKA-II), and major histological differentiation, did not show any significant differences (Supplementary Table 1).

### Investigating HCC heterogeneity using a dual gradient chip

Unlike other types of cancers, HCC exhibits heterogeneous characteristics due to its dual blood supply from the hepatic artery (O$_2$ ↑) and portal vein (O$_2$ ↓) to the liver. This results in variability among patients and within the tumor mass (Fig. 1a)[20]. The arterial phase of the MRI sequence was utilized to distinguish between the two types of HCC. The typical HCC, which receives an abundant arterial supply (O$_2$ ↑), displayed high signal intensity on the MRI. Conversely, the IRE type, with reduced arterial supply (O$_2$ ↓), exhibited low intensity. The difference in oxygen levels significantly affects drug responsiveness, with IRE HCC demonstrating resistance to current drugs and higher recurrence rates, while the typical HCC remains more susceptible to existing treatment approaches. To investigate these oxygen preferences, an oxygen gradient chip for culturing HCC tissue fragments was developed. This chip facilitated the simultaneous perfusion of normoxic and hypoxic media through separate inlets (Fig. 1b), enabling the sorting and collection of HCC cells based on their oxygen preferences. The chip results were further cross-validated by modifying the PDX model, where microchannel chips containing the same patient tissues were implanted in different locations in mice: the normal limb (normoxia), subcutaneous spot (sub-hypoxia), and ischemic hindlimb (hypoxia). Afterward, a drug gradient was established by

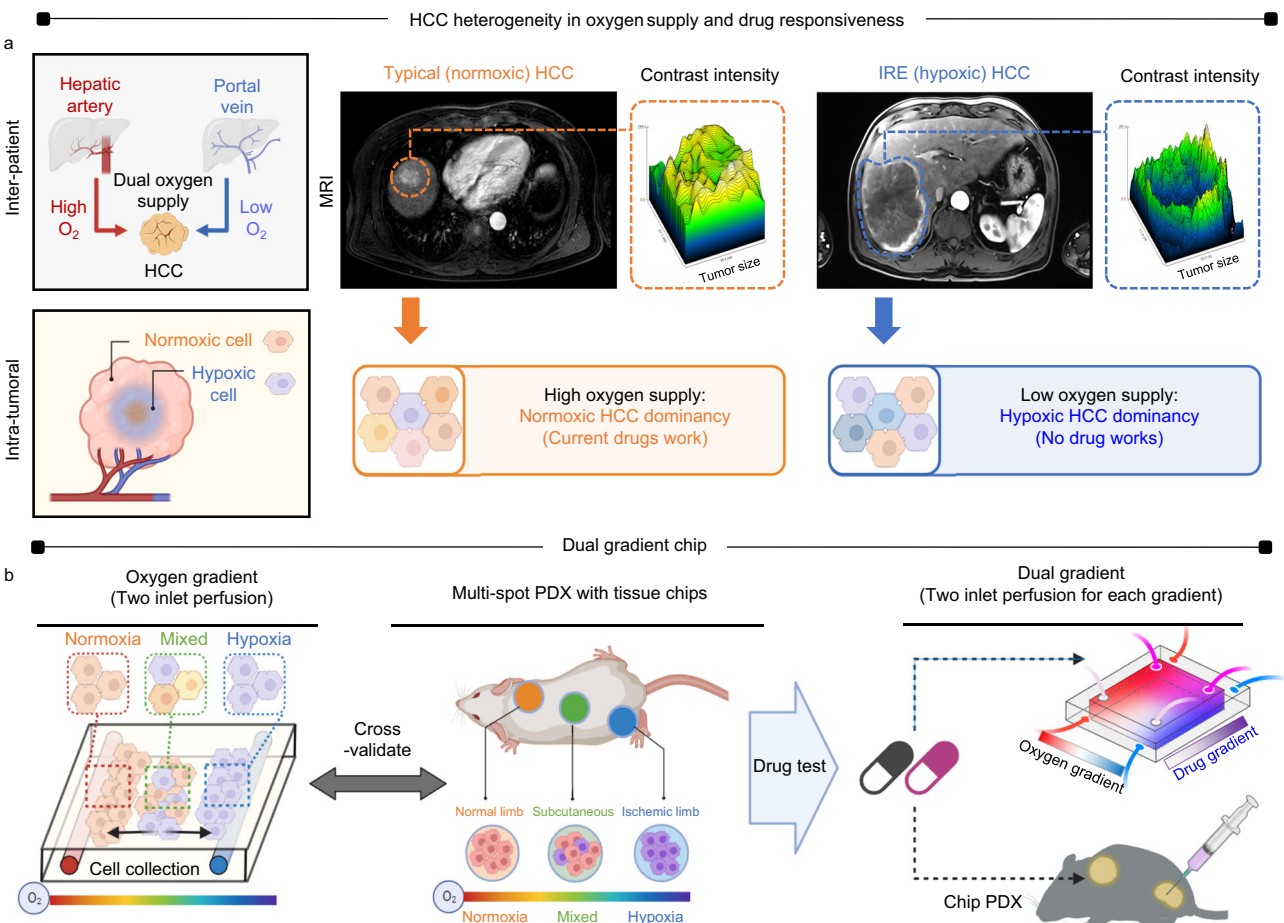

**Fig. 1 | Dual gradient chip as a tool for elucidating hepatocellular carcinoma (HCC) heterogeneity. a** HCC is exposed to a unique microenvironment characterized by a dual blood supply from the hepatic artery ($O_2$ ↑) and portal vein ($O_2$ ↓), which leads to inherent inter-patient and intra-tumoral heterogeneities. During the arterial phase of the MRI sequence, typical HCC ($O_2$ ↑) exhibits high signal intensity due to its abundant arterial supply, in contrast to IRE HCC ($O_2$ ↓), which shows lower signal intensity attributed to reduced arterial supply. Consequently, while typical HCC responds to current drug treatments, IRE HCC often displays resistance to available therapies. **b** The chip is designed to perfuse normoxic and hypoxic media through two separate inlets, facilitating the separation and collection of HCC cells based on their distinct oxygen preferences. Patient tissue obtained through surgery is cultured directly on the chip. Results are then cross-validated by implanting the same tissue-containing chips into normoxic normal limbs, mixed subcutaneous spots, and hypoxic hindlimbs in mice, thus modifying the patient-derived xenograft (PDX) model. Subsequently, a dual perfusion system is established with drug concentration set perpendicular to oxygen levels. This enables simultaneous drug screening alongside HCC separation. Finally, the results undergo further cross-validation using mouse models through drug injection.

perfusing media with and without drugs (Doxorubicin or Sorafenib) through inlets perpendicular to the oxygen gradient. This dual-chip setup allowed for drug screening while separating HCC cells. Furthermore, the chip PDX model in mice was utilized to cross-validate the results by administering drugs orally (Sorafenib) or via injection into the tail vein (Doxorubicin).

### Establishing chip oxygen gradients and determining the diffusion coefficient

The oxygen gradient chip was created by embedding microchannel networks within an enzyme-cross-linkable gelatin hydrogel. This hydrogel contained two pairs of inlets and outlets, allowing for the simultaneous perfusion of normoxic and hypoxic media in parallel (Fig. 2a). Clumps of Soluplus® fibers were incorporated into the hydrogel solution and cross-linked using microbial transglutaminase (mTG) above the low critical solution temperature (LCST) of 38 °C. At RT, the fibers transitioned from a gel to a solution state, creating a network of void microchannels upon removal of residual fibers via PBS perfusion. The density of the fibers was adjusted to optimize the facilitation of oxygen diffusion. This resulted in a dual gradient chip with two pairs of inlets and outlets, enabling the perpendicular

perfusion of oxygen and drugs, which served as the basis for CFD modeling and gradient validation. In contrast, a single-inlet chip was utilized to investigate the effects of fiber density. The chip was perfused solely with normoxic media ($O_2$ concentration = 5.57 mL$^{-1}$) to maximize the effects of channel density and allow for controlled regulation of oxygen transport by adjusting microchannel-generated porosity (Supplementary Fig. 2a). The figure depicts the setup for establishing the oxygen gradient within the hydrogel (sky blue), with the hypoxic area marked by the border (blue). Oxygen levels were measured along the yellow line at incremental distances from the inlet using a probe after 1 min of media perfusion. As fiber density increased from 0 to 3 mg mL$^{-1}$, the number of microchannels also increased, as confirmed by hydrogel porosity (%) (Fig. 2b). In this setup, the diffusion coefficient (D) could be determined at each measurement point along the incremental distance from the single inlet, independent of the effects of two-inlet perfusion. After 1 min of perfusion, increasing fiber density in the single-inlet chip led to a significant increase in oxygen concentration at the dotted line (red), located 10 mm away from the inlet (orange arrow) (Fig. 2c). As microchannel density increased from 0 to 3 mg mL$^{-1}$, overall oxygen diffusion also increased at the measurement points. Moreover, the linearity of the decremental

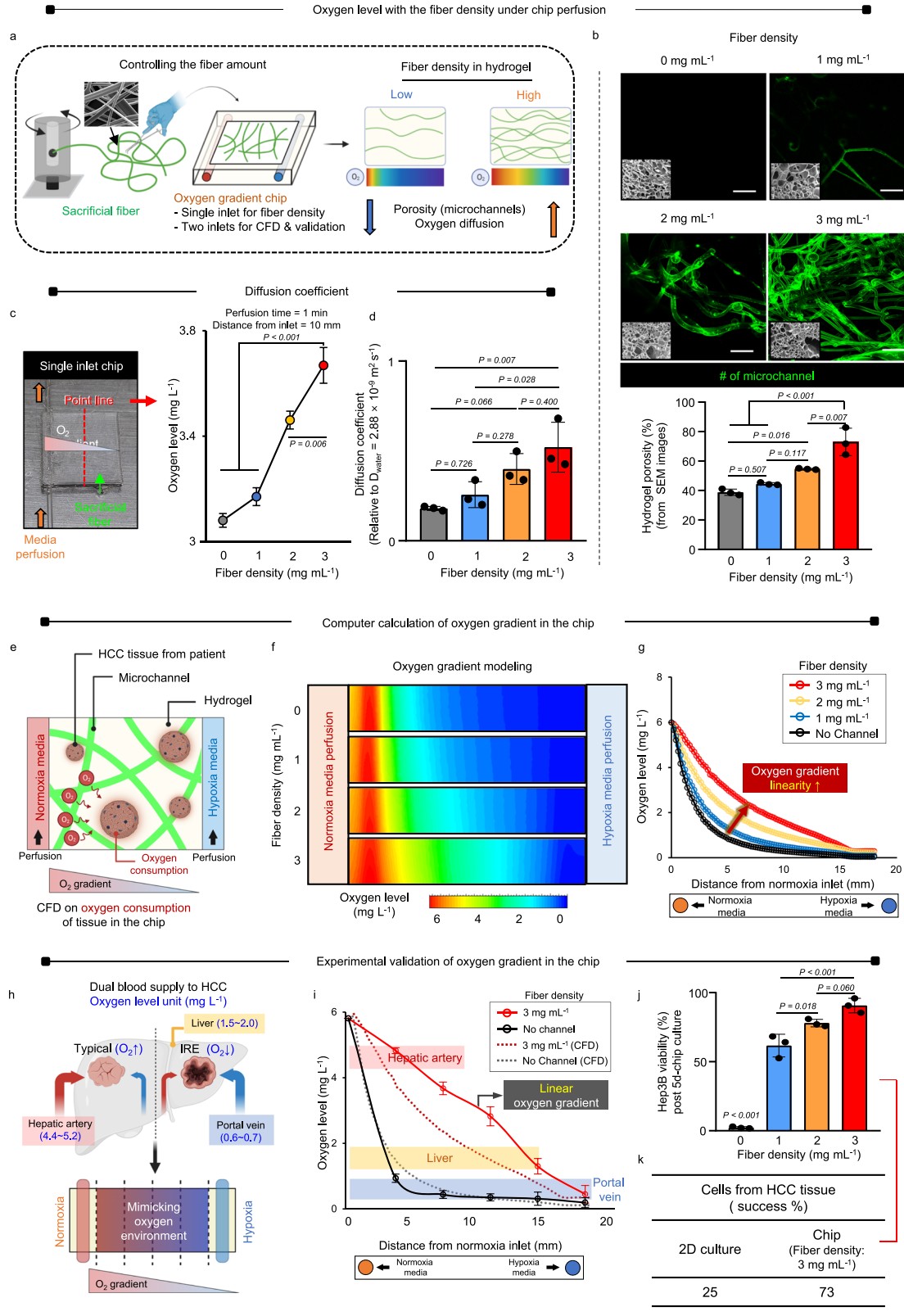

slope improved with increasing distance from the inlet (Supplementary Fig. 2b). Assuming equal D between the media and oxygen upon perfusion, color-contour variations throughout the chip hydrogel were modeled using CFD (Supplementary Fig. 2c). The diffusion coefficient of oxygen in the media ($D_{OM}$) for oxygen-carrying media (water) at 37 °C decreased as the $D_{OM}$ ratio decreased from 1 (1 $D_{OM} = 2.88 \times 10^{-9} \text{ m}^2\text{ s}^{-1}$) to 0.5, 0.2, 0.1, and further to 0.05

(Supplementary Fig. 2d). Consequently, changes in the color contour indicated alterations in the gradient pattern, with the most significant decreases in oxygen level observed at 0.05 $D_{OM}$ along the incremental distance from the inlet. These CFD results were then utilized to calculate oxygen levels at incremental distances from the inlet relative to the $D_{OM}$ ratio (Supplementary Fig. 2e). The analysis confirmed that the most significant decrease in oxygen levels occurred at 0.05 $D_{water}$, with

**Fig. 2 | Experimental setup, modeling, and validation of the chip oxygen gradient. a** Manufacturing the chip and controlling fiber density to regulate oxygen diffusion within the hydrogel. **b** The fiber density increases from 0 to 3 mg mL$^{-1}$ in the single-inlet chip under normoxic media perfusion, the number of microchannels also increases, as confirmed by fluorescence staining. This is supported by scanning electron microscope (SEM) hydrogel porosity analysis (Data are shown as mean ± SD, $n = 3$ biological replicates, Scale bars = 100 μm). **c** When the fiber density is increased in the single-inlet chip, the oxygen level increases at the point line (red) located 10 mm away from the inlet (orange) after 1 min of perfusion. Data are shown as mean ± SD, $n = 3$ biological replicates. **d** The hydrogel's diffusion coefficient (D) shows a stepwise or increase with higher fiber density. Data are shown as mean ± SD, $n = 3$ biological replicates. **e** The oxygen gradient is computed using computational fluid dynamics (CFD) modeling on the two-inlet chips perfused with normoxic and hypoxic media. **f** The oxygen gradients are depicted using color-contour diagrams once CFD simulation reaches steady-state, and **g** linearity from hypoxia to normoxia, is enhanced by increasing the fiber density from 0 to 3 mg mL$^{-1}$. **h.** The computational calculation is carried out in the two-inlet chip system to simulate location-dependent oxygen levels in the liver, hepatic artery, and portal vein. **i** The oxygen concentration in the chip ($5 \times 10^6$ cells per chip) improves linearity with fiber density from 0 to 3 mg mL$^{-1}$ (solid lines), consistent with computational modeling (dotted lines). Data are shown as mean ± SD, $n = 3$ biological replicates. **j** Cell viability increases at 3 mg mL$^{-1}$ fiber density. Data are shown as mean ± SD, $n = 3$ biological replicates. **k** The success of HCC tissue culture is defined by the production of viable cells capable of passaging. Significance was determined using one-way ANOVA with Tukey's test versus between lined groups. Source data are provided as a Source Data file.

the lowest level of oxygen concentration recorded at the measurement point. Oxygen levels (y-axis, mg L$^{-1}$) were plotted against the $D_{OM}$ ratio (x-axis, relative to 1 $D_{OM} = 2.88 \times 10^{-9}$ m$^2$ s$^{-1}$) based on CFD calculations (Supplementary Fig. 2f), and the probe measurements were integrated into the y-axis of the CFD calculations. Consequently, actual D values were obtained as a function of the fiber density in the two-inlet chip under simultaneous perfusion of normoxic and hypoxic media. As fiber density increased, there was a corresponding stepwise and significant increase in the diffusion coefficient of the hydrogel (Fig. 2d).

## Two-inlet CFD modeling with experimental validation

The oxygen gradient in the CFD model of the two-inlet chip was established by controlling the media supply (oxygen concentration and flow rate) and diffusion (D and permeability of hydrogel) as boundary conditions (Supplementary Fig. 3a). Additionally, the oxygen consumption rate of HCC cells played a crucial role in determining the gradient. By performing step-by-step calculations of (i) the number of cells in the HCC tissue, (ii) the average number of cells in the chip, and (iii) the oxygen consumption rate per cell, the oxygen consumption rate in the chip was determined to be $1.6 \times 10^{-2}$ μg s$^{-1}$ (Supplementary Fig. 3b). This value represented the oxygen consumption by HCC cells, and the influx volume of hypoxic oxygen was calculated (0.007 mL s$^{-1}$) for input into CFD modeling (Supplementary Fig. 3c). The oxygen levels in both the normoxic and hypoxic media remained stable during the perfusion culture (Supplementary Fig. 4). In particular, the hypoxic medium consistently maintained the oxygen concentration below 0.35 mg L$^{-1}$ before media change, ensuring the hypoxic environment. This hypoxic oxygen level is lower than the physiological range in the portal vein (0.6-0.7 mg L$^{-1}$) although there was a gradual increase in the overall trend of oxygen level, suggesting the need of media exchange every 24 h. CFD modeling was used to simulate the oxygen gradient of the two-inlet chip. Normoxic and hypoxic media were simultaneously perfused, with varying fiber density (Fig. 2e). The finite element analysis model considered the diffusion coefficient, two media supplies, and the oxygen consumption rate of patient tissue under chip culture as boundary conditions. After performing computational calculations, the oxygen gradients were visually represented as color-contour diagrams from the CFD simulation at steady-state. Increasing the fiber density facilitated more effective oxygen diffusion throughout the chip, as indicated by the gradual change in colors (Fig. 2f). The linearity of the observed oxygen gradient, from hypoxia to normoxia, with increasing fiber density (0 to 3 mg mL$^{-1}$) further confirmed the accuracy of the CFD results (Fig. 2g). The computational analysis conducted in the two-inlet chip system simulated the levels of oxygen in different locations within the liver: the hepatic artery (4.4-5.2 mg mL$^{-1}$), the portal vein (0.6-0.7 mg mL$^{-1}$), and the liver itself (1.5-2.0 mg mL$^{-1}$) (Fig. 2h). To experimentally validate these simulations, oxygen levels were measured on day 5 post-perfusion of normoxic and hypoxic media through each inlet in the chip under Hep3B culture conditions ($5 \times 10^6$ cells per chip). The results showed an increase in the linearity of the oxygen gradient as the

fiber density increased from 0 to 3 mg mL$^{-1}$, consistent with the results of the CFD model. Additionally, the two-inlet chip was validated by showing three distinct ranges of oxygen levels from the liver on the graph (Fig. 2i). The viability of Hep3B cells significantly increased with increasing fiber density, as assessed by the CCK-8 assay and Live/Dead staining (Fig. 2j, Supplementary Fig. 5). Notably, the group without fibers exhibited significant cell death, highlighting the effectiveness of the chosen fiber density of 3 mg mL$^{-1}$. Furthermore, successful HCC tissue culture was defined by the presence of viable cells that could be passed. Remarkably, the chip with a fiber density of 3 mg mL$^{-1}$ demonstrated a success rate of 73% compared to 25% for 2D culture, further confirming the efficiency of chip culture (Fig. 2k).

## Hypoxic IRE HCCs present challenges in the current HCC clinic

During the arterial phase of MRI imaging (Fig. 3a), typical HCC (patient #1, and #7 normoxia) exhibited contrast enhancement, indicating a preference for high levels of oxygen. In contrast, IRE HCC (patients #3 and #8, hypoxia) displayed hypoenhanced contrast due to its lower affinity for oxygen, highlighting the heterogeneity among patients. When HCC patient tissues were cultured on the oxygen gradient chip for 4 days (Fig. 3b, Supplementary Fig. 6), typical HCC showed increased expression of protein markers associated with hypoxia-mediated stemness (CAIX), cancer invasiveness (K19), and microvascular invasion (CD34) on the normoxic side compared to the hypoxic side. Conversely, IRE HCC exhibited increased marker expression in the hypoxic zone relative to the normoxic zone, confirming that the chip effectively preserved the inherent tissue characteristics related to oxygen preference. After the 4-day culture on the chip, the TUNEL assay results confirmed the MRI findings, with typical HCCs demonstrating higher viability on the normoxic side compared to the hypoxia-preserved viability of IRE HCCs (Fig. 3c). This was supported by a larger number of nuclei in both typical and IRE HCCs on the normoxia and hypoxia sides, respectively, compared to the opposite sides (Supplementary Fig. 7). Furthermore, intra-tumoral heterogeneity in IRE HCCs became evident, with viable cells found in both the hypoxic (95%) and normoxic (45%) regions.

## On-chip separation and collection of HCC cells based on oxygen levels

The ability of the chip to separate and collect HCC cells was demonstrated by culturing a mixture of typical (patient #6, high MRI contrast) and IRE HCC cells (patient #9, low MRI contrast) in the chip (Fig. 4a). For 7 days, the cells were simultaneously perfused with normoxic (20% O$_2$) and hypoxic (0.1% O$_2$) media through the two inlets. Confocal imaging revealed that the IRE cells (red DiI) gradually migrated to the hypoxic side, while the typical cells (green DiO) remained on the normoxic side with Zen 3.3 blue edition. The chip effectively separated and collected the two types of HCC cells on their respective sides, as indicated by the day-7 histogram (Supplementary Fig. 8a). Based on their oxygen preferences, after 3 days of 2D culture, the proliferation of typical HCC cells was higher in normoxia compared to hypoxia.

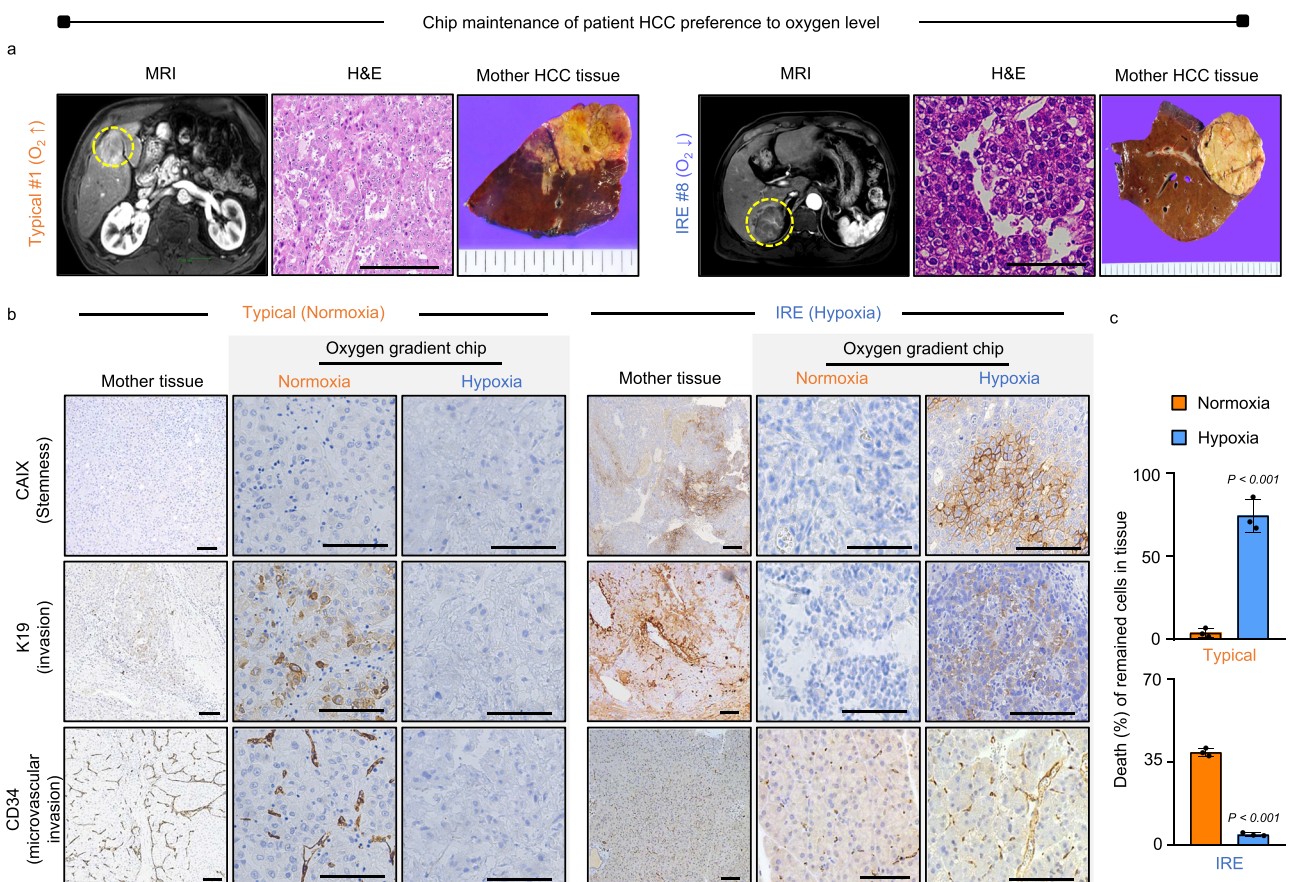

**Fig. 3 | Hypoxic IRE HCCs as a challenge in the current HCC clinic. a** During the arterial phase of MRI imaging, typical HCC (patient #1, normoxia) demonstrates contrast enhancement (yellow circle), indicating a high preference for oxygen, while IRE HCC (patient #8, hypoxia) exhibits hypoenhanced contrast (yellow circle) due to a low preference for oxygen, indicating intra-tumoral heterogeneity ($n = 3$ biopsy specimens from each patient, Scale bars = 200 µm). **b** Immunohistochemistry analyses of both typical and IRE HCCs reveal increased expression of protein markers associated with hypoxia-mediated stemness (CAIX), cancer invasiveness (K19), and microvascular invasion (CD34) on both the normoxia and hypoxia sides, compared to the opposite side (Scale bar = 200 µm). **c** The patient tissues are cultured in the oxygen gradient chip for four days and then subjected to a TUNEL assay to analyze the death (%) of remained cells in the sample tissue. Consistent with the MRI findings, typical HCC shows higher variability on the normoxic side, while IRE HCC exhibits preserved viability under hypoxic conditions. Data are shown as mean ± SD, $n = 3$ biological replicates in each patient. Significance was determined using a two-sided t-test without adjustments for multiple comparisons versus between normoxia. Source data are provided as a Source Data file.

Conversely, hypoxia significantly promoted the growth of IRE cells (Fig. 4b). On day 7 (Fig. 4c), marker gene expression related to hypoxia (*CAIX*, *VEGF*, *GLUT1*) and drug resistance (*ABCB1*, *ABCC2*, *GSTP1*) increased on the hypoxic side of the chip compared to day 4, while the expression levels on the normoxic side generally decreased. The differential expression patterns became more uniform on both sides, indicating the progressive separation of HCC types (Typical: #4, #5, #6 and IRE: #7, #8, #9) in preparation for collection. The cell death was promoted when the cells remained in the tissues without migration to the preferred oxygen condition (Fig. 3c). However, when a mixture of typical and IRE (#4 + #7 or #5 + #8) was cultured without tissues as a free migration condition (Supplementary Fig. 8b), the cell viability did not decrease from day 0 to 4 and further to 7 upon live and dead assays. The results indicate the cell migration to the HCC-type-specific oxygen condition for survival (Fig. 4a) with the consequent high rate of proliferation (Fig. 4b) was not affected by the death rate when the cell migration was not hindered by remaining in the tissue (Fig. 3c). In contrast, no clear separation of cells was resulted under uniform normoxia or uniform hypoxia without the gradient, indicating a key role of oxygen gradient in separation of HCC-type-specific cells into their preferred oxygen condition (Supplementary Fig. 9). This highlights the limitations of conventional uniform culture systems in separating cells that contribute to recurrence, thereby preserving

heterogeneity. When patient-derived cells were cultured in the chip for 10 days, the phenotypic gene profiles were maintained with respect to hypoxia (*CAIX*, *VEGFR*, *GLUT1*), drug resistance (*ABCB1*, *ABCC2*, *GSTP1*), ECM remodeling (*CD44*, *Vim*, *MMP9*), and HCC indicators (*AFP*) in the normoxic and hypoxic zones. Typical (#7) cells retained the genetic profile of mother tissue in the normoxic zone in contrast to the hypoxia-specific retention of IRE (#9) cells (Supplementary Fig. 10), indicating stable phenotypic maintenance of patient-derived cells upon chip culture for the experimental periods. Overall, the expression patterns on both sides became more homogeneous, indicating a progressive separation of HCC types in preparation for collection. These findings were further supported by increased expression of marker proteins associated with hypoxia (*CAIX*) and drug resistance (*ABCB1*) in hypoxia compared to normoxia (Fig. 4d).

## Dual gradient chip for hypoxia-targeted collection of drug-resistant IRE cells
After a 7-day culture of HCC patient tissue (#3, #8, #9) using a two-inlet perfusion system, the oxygen gradient chip was able to separate drug-resistant IRE cells to the hypoxic side. The protein marker expression of microvascular invasion (CD34) and stemness (CAIX) across the chip provided evidence of the separation (Fig. 5a, Supplementary Fig. 11). Furthermore, upon chip collection, quantitative zymography of

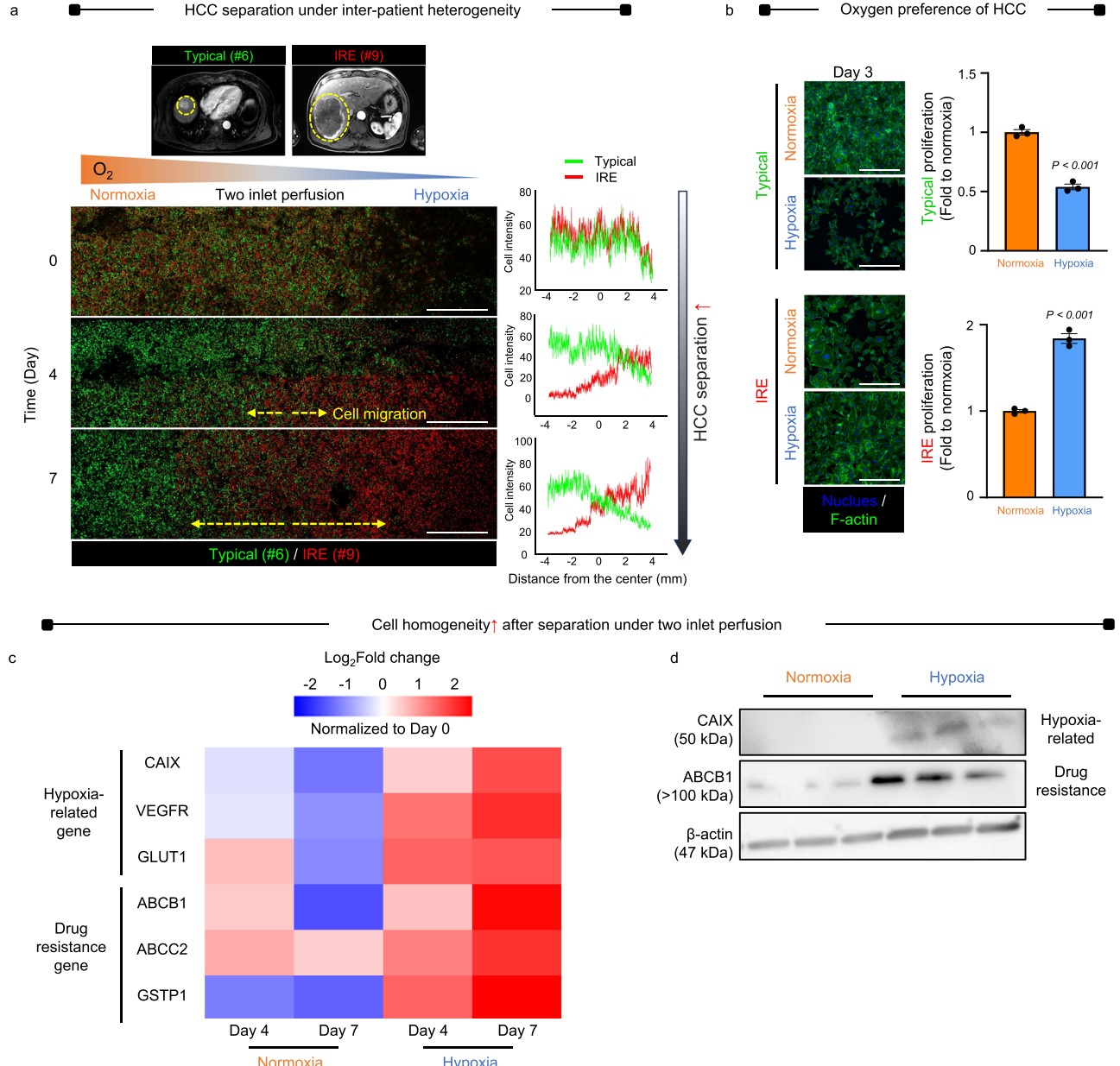

**Fig. 4 | On-chip separation and collection of HCC cells based on oxygen levels.** **a** A mixture of typical HCC cells (patient #6, high MRI contrast: yellow circle) and IRE HCC cells (patient #9, low MRI contrast: yellow circle) is cultured in the chip under two-inlet perfusion of normoxic and hypoxic media for 7 days. Confocal imaging shows that over time, IRE cells (red DiI) progressively migrate to the hypoxic side, while the typical cells (green DiO) remain predominantly on the normoxic side. The two types of HCC cells are separated and collected on the normoxic and hypoxic sides, respectively, as shown in the day-7 histogram ($n = 3$, Scale bar = 1 mm). **b** After 3 days of 2D culture, typical HCC cells exhibit increased proliferation in normoxic conditions compared to hypoxic environments. In contrast, hypoxia significantly enhances the growth of IRE HCC cells (Scale bare = 100 μm). Data are shown as mean ± SD, $n = 3$ biological replicates. Significance was determined using a two-sided t-test without adjustments for multiple comparisons versus between normoxia. **c** On day 7, marker gene expression related to hypoxia (*CAIX*, *VEGF*, *GLUT1*) and drug resistance (*ABCB1*, *ABCC2*, *GSTP1*) increases on the hypoxic side of the chip compared to day 4, while the expression levels on the normoxic side generally decrease. The differential expression patterns become more uniform on both sides, indicating the progressive separation of HCC types (Typical: #4, #5, #6 and IRE: #7, #8, #9) in preparation for collection. The results are presented as Log$_2$Fold change values with normalization to day 0, which are visualized using the heatmap function in RStudio ($n = 3$ biological replicates). **d** These findings are further supported by elevated marker protein expression of hypoxia (CAIX) and drug resistance (ABCB1) in hypoxic conditions compared to normoxia. The samples derived from the same experiment and that gels were processed in parallel. Source data are provided as a Source Data file.

proteinase (MMP1) activity confirmed the invasive nature of IRE HCC cells (Fig. 5b). The migration of invasive IRE cells toward the hypoxic side necessitates ECM remodeling to guide cell movement. This was accomplished by augmenting MMP1 activity, which was monitored at 4-min intervals over 90 min. Using the dual gradient chip, IRE-dominant HCC tissues were cultured under perpendicular oxygen and drug gradients, with two-inlet perfusions for each gradient (Fig. 5c). The

characteristics of drug-resistant HCC cells were evaluated along the oxygen gradient. In the experiment, HCC tissues harvested immediately after surgery were embedded into the chip and cultured for 7 days solely under the influence of the oxygen gradient. Subsequently, the cells were exposed to a drug gradient set perpendicular to the oxygen gradient for an additional 3 days. Finally, a live and dead assay was conducted on day 10 (Fig. 5d). IRE-dominant HCC tissues from patients

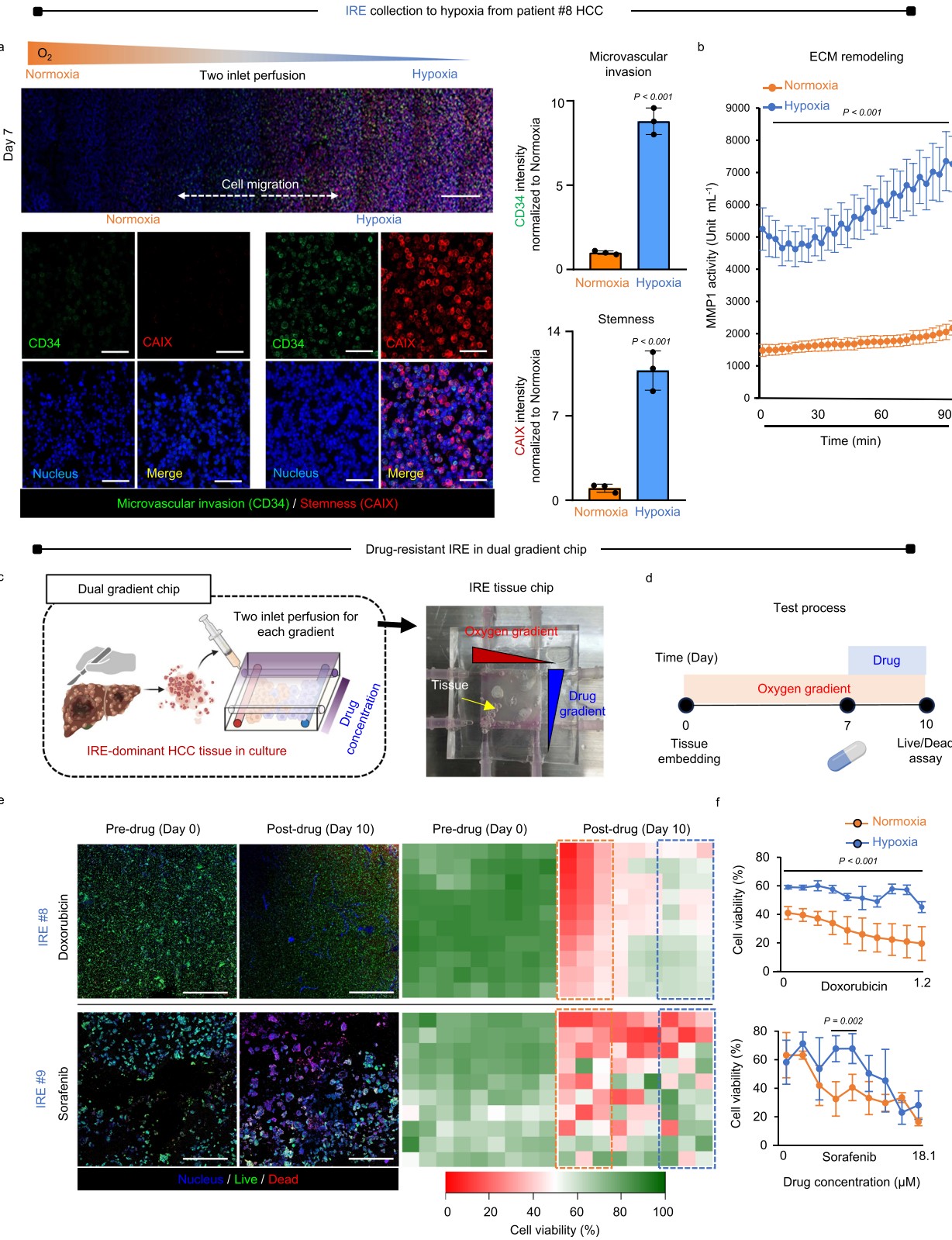

#8 and #9 were then subjected to gradients of doxorubicin (top) and sorafenib (bottom), as shown in Fig. 5e. In both instances, a decrease in drug concentrations led to increased cell viability due to increased hypoxia. Similarly, patient #9 exhibited resistance to sorafenib, a widely used multi-tyrosine kinase inhibitor for stage 4 HCC, specifically within the hypoxic zone. These clinical outcomes substantiate chip results.

Moreover, quantitative analyses indicate that even when doxorubicin concentrations were increased to 1.2 μM, patient #8's clinical outcome did not improve. Conversely, increasing the dosage of sorafenib could potentially improve patient #9's likelihood of survival (Fig. 5f). These results affirm the practical value of the chip in predicting therapeutic outcomes within a clinical context.

**Fig. 5 | Dual gradient chip for the collection of drug-resistant IRE cells into hypoxia from HCC tissue cultures. a** The oxygen gradient chip enables the collection of drug-resistant IRE cells towards the hypoxic side after culturing HCC patient tissue (#8) for 7 days under dual inlet perfusion. This is evidenced by the expression of protein markers associated with microvascular invasion (CD34) and stemness (CAIX) across the entire chip (top, scale bar = 1 mm). Immunostaining with magnification (bottom) and subsequent quantitative image analysis (right) confirm the presence of these markers (scale bar = 100 μm). Data are shown as mean ± SD, $n = 3$ biological replicates. **b** Zymographic quantitative analysis of proteinase (MMP1) activity further confirms the invasive traits of IRE HCC cells upon chip collection. This activity was quantified through a 90-min monitoring period with 4-min intervals. Data are shown as mean ± SD, $n = 3$ biological replicates. **c** The dual gradient chip is used to culture IRE-dominant HCC tissues under perpendicular gradients of oxygen and drugs, facilitated by two-inlet perfusions for each gradient. **d** HCC tissues are cultured on the chip under oxygen gradients for 7 days, followed by drug exposure for an additional 3 days, with a live and dead assay conducted on day 10. **e** The IRE-dominant HCC tissues from patients #8 and #9 are exposed to gradients of doxorubicin (top) and sorafenib (bottom), respectively. In both cases, viability increases as hypoxia intensifies under decreasing drug concentrations, as observed in fluorescence imaging with 2D heatmap analyses (Scale bar = 200 μm). **f** Patient #9 showed resistance to sorafenib within the hypoxic zone. High doses of sorafenib could potentially enhance the survival prospects of patient #9, highlighting the chip's efficacy in predicting therapeutic strategies in the clinical setting. Data are shown as mean ± SD, $n = 3$ biological replicates. Source data are provided as a Source Data file.

## Multi-spot chip PDX model as an in vivo avatar of the chip

Oxygen levels progressively increase from the ischemic limb (IL: hypoxia) to the subcutaneous spot (SC: mixed, sub-hypoxia), and ultimately to the normal limb (NL: normoxia) due to corresponding incremental vascularization in mice (Fig. 6a). These physiological conditions were designed to reflect changes in the oxygen gradient observed in the chip. HCC tissue-containing chips were implanted into the three locations in mice for 2 weeks, followed by 4 weeks of drug injections. Histological analysis was conducted at 6 weeks. This mouse model was referred to as the multi-spot chip PDX model. Oxygen levels in NL (4.69 ± 0.43 mg L$^{-1}$), SC (2.51 ± 0.12 mg L$^{-1}$), and IL (0.66 ± 0.07 mg L$^{-1}$) measured using an oxygen probe correlate with the oxygen levels of the hepatic artery (4.4–5.1 mg L-1), liver (1.5–2 mg L$^{-1}$), and portal vein (0.6–0.7 mg L$^{-1}$), respectively, validating the use of the model (Fig. 6b). Vascularization also increases from IL to SC, then to NL, as indicated by quantifying CD31 expression through immunohistochemistry and H&E examination (Fig. 6c). Patient #8, exhibited poor arterial contrast on MRI, indicating IRE dominance. This was confirmed through histological examination of the original tissue using H&E staining, which revealed heterogeneous expression of the stemness marker (CAIX) (Fig. 6d). Subsequently, the tissue-containing chips were implanted into mice and monitored for 6 weeks, resulting in histological changes consistent with IRE HCC. There was a gradual increase in CAIX expression from NL to SC, and then to IL tissue. However, the expression of the HCC indicator (AFP) remained intact, indicating the utility of the multi-spot PDX model in preserving HCC characteristics. Moreover, when the tissue from patient #8 was implanted under IL-induced hypoxia, intrahepatic metastasis occurred (Supplementary Fig. 12). The expression of marker genes associated with hypoxia (*CAIX, VEGFR*, and *GLUT1*), drug resistance (*ABCB1, ABCC2*, and *GSTP1*), and ECM remodeling (*CD44, Vim*, and *MMP9*) progressively increased from NL to SC, and further to IL (Fig. 6e), suggesting a higher likelihood of drug-resistant recurrence following TACE therapy. The evaluation of drug response in the chip multi-spot PDX model over 6 weeks (Fig. 6f) demonstrated significant reductions in tumor volume over time with both doxorubicin and sorafenib in NL and SC, but not in IL, thus confirming the drug resistance of IRE HCC in IL-induced hypoxia.

## Discussion

Advances in precision medicine for hepatocellular carcinoma aim to personalize drug treatments. However, challenges persist due to tumor complexities and patient heterogeneities. One major source of heterogeneity in the liver comes from location-dependent variations in oxygen levels that result from the simultaneous blood supply via the oxygen-rich hepatic artery and oxygen-deficient portal vein. This heterogeneity significantly contributes to low engraftment rates in conventional culture and PDX models[39]. The liver is comprised of lobules as a form of hexagonal units, where three zones are divided following the degree of oxygen supply. The periportal zone 1 surrounds the portal tracts and receives oxygenated blood from the hepatic arteries.

On the other hand, the perivenous zone 3 is located adjacent to the central veins, where the oxygen supply is limited. Blood flows from the portal tract to the venules through the three zones, creating an oxygen gradient within the liver. Simulation of the hypoxic environment of perivenous zone 3 (Fig. 3) enables the detection and isolation of hepatic cells with carcinogenic potential[40,41], offering insights into HCC progression and therapeutic strategies. This heterogeneity significantly contributes to the low engraftment rates in the conventional culture and PDX models. In light of these considerations, the study proposed a clinic-aided method to predict patient-specific prognosis. This method involves surgically removing HCC tissues and culturing them on a dual gradient chip. The chip acts as an avatar of the dual supply, mixing arterial and venous blood in varying location-dependent ratios. Computational modeling and experimental validation were integrated with microchannel functionality to improve the linearity of the oxygen gradient, ensuring an accurate representation of physiological oxygen diffusion. The method was also connected to an in vivo model by implanting HCC tissue into normoxic, sub-hypoxic, and hypoxic sites in mice. This was done to account for systemic variations in oxygen levels. The chip was designed to replicate the natural gradients of oxygen, pH, and chemical concentration in the body, which are essential for biological processes. Among these, environmental oxygen levels play a key role in regulating cancer stemness and drug resistance. Cells not only respond to specific oxygen concentrations, but also to gradients, which can influence cancer behaviors such as migration, invasion, and metastasis[24–26]. Key parameters of oxygen diffusion were calculated using CFD, taking into account factors like cellular oxygen consumption, rheological conditions, and diffusion changes in the 3D culture system. The microchannel networks within the chip serve as interactive and regulatory pathways between the oxygen gradient and HCC cells. The controlled channel density facilitates efficient oxygen diffusion, providing a framework for understanding the migratory tendencies of HCC cells toward their preferred oxygen levels within the chip. As a result, the chip significantly improves engraftment rates and is able to reflect the inter-patient heterogeneity observed in clinical MRI findings. The sustained viability of cells and preservation of HCC characteristics suggest the chip's potential to overcome challenges faced by traditional avatar models. During 10 days in the chip culture (Supplementary Fig. 10), patient-derived cells underwent expression changes in the phenotypic genes that are associated with hypoxia, drug resistance, ECM remodeling, and HCC markers. Typical (#7) and IRE (#9) cells retained the mother tissue-like profiles in the normoxic and hypoxic zones, respectively, indicating the stable maintenance of cell characteristics by the chip for the experimental period. This claim was supported by the 10-day drug responses of patient tissues in the chip (Fig. 5) and the multi-spot chip PDX model for over 6-week period of implantation as the IRE and typical characteristics were maintained when the histological features, gene profiling, and drug responses were analyzed (Fig. 6). Moreover, our previous study verified that direct culture of patient-derived tissue in the chip system with the absence of oxygen gradient resulted in

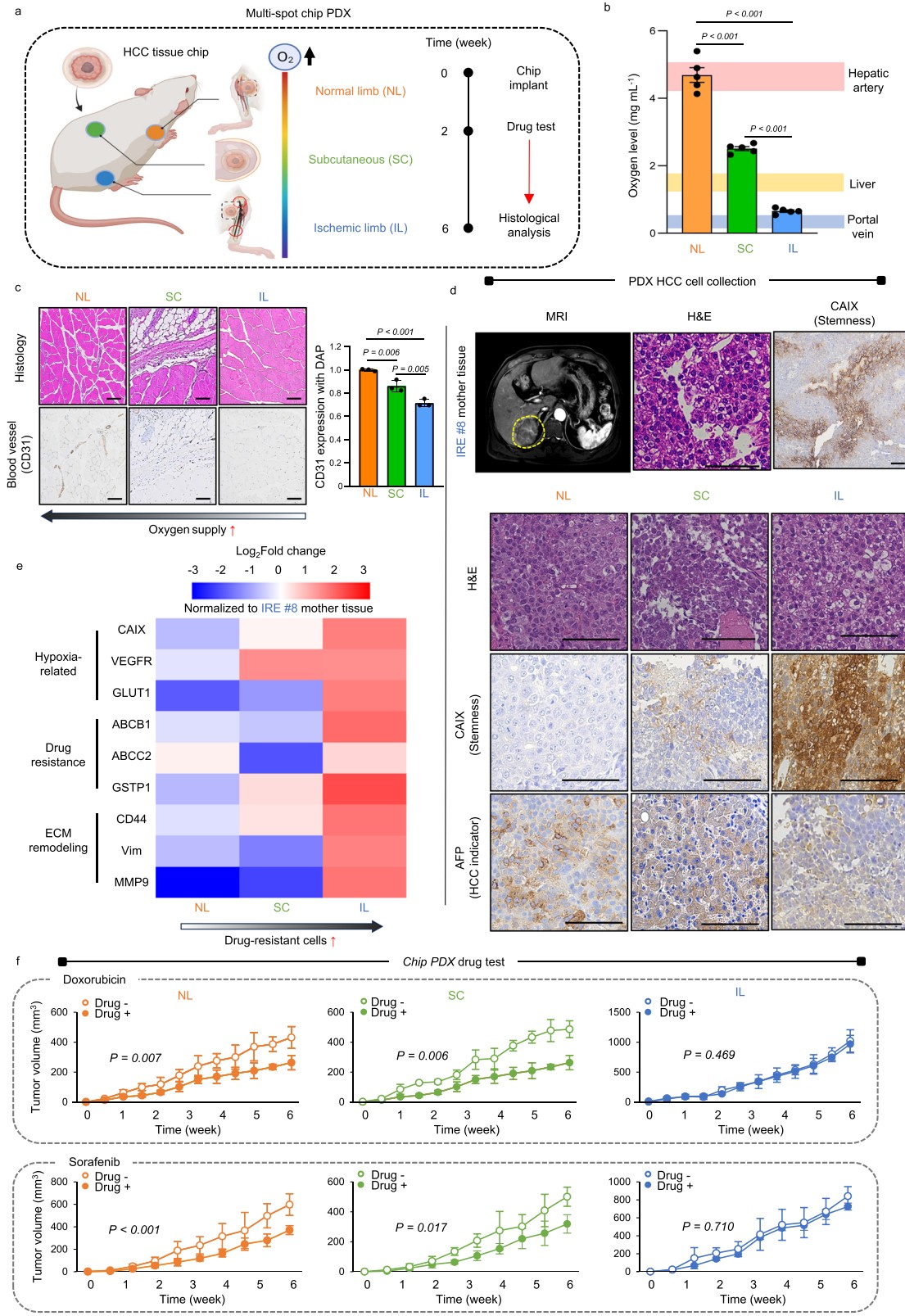

stable maintenance of the genetic characteristics in patient tissues over a 3-week period[42]. The oxygen gradient function, coupled with drug concentration gradients, has emerged as a reliable predictor of clinical outcomes and a valuable tool for guiding treatment strategies. Over the past three years, tissue samples from 12 early-stage HCC patients were collected during surgery. These samples were analyzed and compared with the results obtained from the chip. The poor prognosis observed in IRE HCC patients, which is attributed to recurrence due to the treatment resistance in hypoxic environments, corresponds with the increased drug resistance observed in the dual gradient chip. This correlation was further validated by cross-validation with an in vivo chip PDX model, which utilized different anatomical locations in mice as bioreactors to analyze cancer characteristics across varying oxygen and drug concentrations. Through

**Fig. 6 | Multi-spot chip PDX model as an in vivo avatar of the chip. a** The chip mimics varying oxygen levels from hypoxia in the ischemic limb (IL) to mixed, sub-hypoxia in the subcutaneous spot (SC), and finally to normoxia in the normal limb (NL). HCC chips implanted for 2 weeks, then drugs given for 4 weeks, and histological analysis at 6 weeks. This mouse model is referred to as the multi-spot chip PDX model. **b** The oxygen levels of NL ($4.69 \pm 0.43$ mg L$^{-1}$), SC ($2.51 \pm 0.12$ mg L$^{-1}$), and IL ($0.66 \pm 0.07$ mg L$^{-1}$) correspond to those of the hepatic artery ($4.4$–$5.1$ mg L$^{-1}$), liver ($1.5$–$2$ mg L$^{-1}$), and portal vein ($0.6$–$0.7$ mg L$^{-1}$), respectively, as measured upon using an oxygen probe. Data are shown as mean $\pm$ SD, $n = 5$ biological replicates. Significance was determined using one-way ANOVA with Tukey's test versus between lined groups. **c** The degree of vascularization (CD31) increased progressively from IL to SC, and further to NL. Protein expression was quantified via DAP intensity alongside H&E examination. Data are shown as mean $\pm$ SD, $n = 5$

biological replicates. Significance was determined using one-way ANOVA with Tukey's test versus between lined groups. **d** The poor arterial contrast (yellow circle) seen on MRI indicates the predominance of IRE in patient #8. The mother tissue exhibits heterogeneous stemness marker (CAIX) ($n = 3$ biopsy specimens from the patient). These mother tissue-containing chips are implanted into three spots in mice for 6 weeks. ($n = 5$ biological replicates). **e** The heatmap analysis of qRT-PCR results displays data as Log2Fold change values normalized to those of the mother tissue ($n = 5$ biological replicates). **f** Drug response was evaluated in the chip multi-spot PDX model over 6 weeks. Data are shown as mean $\pm$ SD, $n = 5$ biological replicates. Significance was determined using a two-sided t-test without adjustments for multiple comparisons versus between without drug. Source data are provided as a Source Data file.

this model, chip results were cross-validated within a physiological system, allowing for the monitoring of intrahepatic metastasis as cancer cells migrated along host vessels following venous drug injection. The proposed methodological platforms encompass computational calculations, chip utilities, and in vivo models, to enhance treatment efficacy and ultimately improve patient outcomes. When combined, these methods offer a reliable guideline for cross-validation across multi-disciplinary fields. However, further research is required to better understand the mechanisms of HCC-type-specific drug resistance for different drugs in the current treatment paradigm. As the small number of HCC cases appeared to be a limitation of the present study, the next study plans to recruit a larger cohort of HCC patients to increase inter-patient heterogeneity and validate the chip approach across various cancer types. Furthermore, the incorporation of a large animal model is considered to enhance the reliability in simulating variations in both oxygen and drug concentrations so that more clinically meaningful data can be generated. In addition, tracking the long-term prognosis alongside clinical screening and evaluation will help refine guidelines and provide a realistic toolkit for assessing the efficacy of drug treatments. Ultimately, integrating these findings with ongoing advancements in organ chip technology for disease modeling and drug screening will contribute to the progress of personalized therapeutic interventions.

## Methods
For the clinical experiments, following approved protocols from the Institutional Review Board (IRB) of Yonsei University College of Medicine (IRB: 4-2016-0728). Animal experiments were conducted in accordance with the guidelines set by the Institutional Animal Care and Use Committee (IACUC) at Yonsei University College of Medicine (#2021-0176).

### Tissue and cell samples from hepatocellular carcinoma (HCC) patients
Twelve patients with HCC consented to donate tissue during surgical tumor removal. Diagnosed in the early stages according to the Barcelona Clinic Liver Cancer system[43], patients were classified into typical and irregular rim-like enhancement (IRE) HCC groups based on medical records and MRI findings. Typical HCC exhibited uniform arterial phase enhancement above the liver's overall level, while IRE HCC displayed irregular enhancement with a bright peripheral border and dim center[20]. The quantitative analysis of MRI contrast intensity was conducted using the segmentation plot plugin from ImageJ (Fiji version 1. 52i, National Institute of Health, MD, USA) along each midsection. Tumors were resected within two months of diagnosis, followed by histological analysis to evaluate tumor characteristics, such as level of differentiation, microvascular invasion, and portal vein invasion. Diagnostic patient information included age, tumor mass size, viral infection status, and tumor marker levels (AFP and PIVKA-II). Regular post-operative monitoring involved CT, MRI, and tumor marker tests to detect recurrence. HCC tissues obtained during surgical resection

were immediately transported to the laboratory in cold culture media. Upon arrival, the tissues were minced using grinder tips (G50; Coyote Bioscience, BJ, China), centrifuged at $1330 \times g$ for 3 min, and weighed. Subsequently, cells were extracted by incubating minced tissue with collagenase type I (210 U mL$^{-1}$, 17100-017; Gibco, Carlsbad, CA, USA) and DNase I (10 µg mL$^{-1}$, 11284932001; Sigma-Aldrich, St. Louis, MO, USA) in Dulbecco's modified Eagle's medium-high glucose with L-glutamine and without sodium pyruvate (DMEM, 11965-092; Gibco), supplemented with 1% (v v$^{-1}$) penicillin-streptomycin (PS, 15140-122; Gibco) for 1 h at 37 °C under continuous shaking. After incubation, the mixture was filtered using a cell strainer with a 70 µm pore diameter (352350; Corning, NY, USA), and collagenase was deactivated by washing the cells three times in DMEM medium with 10% heat inactivated fetal bovine serum (FBS, 16000-044; Gibco). Cells were then cultured in DMEM supplemented with epithelial growth factor (EGF, 20 ng mL$^{-1}$, PHG0311; Thermo Fisher Scientific, Waltham, MA, USA) and basic fibroblast growth factor (b-FGF, 10 ng mL$^{-1}$,100-18C; Pepro-Tech, Rocky Hill, NJ, USA) for 5–10 passages. Additionally, the HCC cell line Hep3B, obtained from the Korean Cell Line Research Foundation (88064; Seoul, Republic of Korea), was cultured in DMEM at 37 °C with 20% O$_2$ and 5% CO$_2$.

### Production of the oxygen gradient chip
For the microchannel networks, microfibers were produced by spinning a PCL-PVAc-PEG copolymer (Soluplus®, 30446233; BASF, Ludwigshafen, Germany) dissolved in a 66% w v$^{-1}$ MeOH solution using a custom device. The diameter of the fibers was adjusted by varying the rotational speed (140–176 g) of the spinning device[42,44–46]. Subsequently, the fibers were placed in a polydimethylsiloxane (PDMS, 31-000810-02; Dow corning, Midland, MI, USA) mold measuring $2 \times 2 \times 0.5$ cm at a density ranging from 0 to 3 mg mL$^{-1}$. The bottom plate of the mold was sealed onto a glass slide (1000412; Marienfeld, Lauda-Königshofen, Germany). Inlet and outlet holes for media perfusion were created by piercing the left and right ends of the PDMS mold with an 18G blunt needle (0721092; Korea Vaccine, Seoul, Republic of Korea). Two pairs of inlets and outlets were arranged in parallel on the upper (normoxia) and lower (hypoxia) sides of the mold to allow simultaneous perfusion of normoxic and hypoxic media through the corresponding inlets. Next, a solution containing gelatin (5.5% w v$^{-1}$, G1890; Sigma-Aldrich) and microbial transglutaminase (1% w v$^{-1}$, mTG, 1201-50; Modernist Pantry LLC, Eliot, ME, USA) in a volume ratio of 9:1 was poured onto the Soluplus® microfibers within the PDMS mold. The mixture was then cross-linked for 30 min at 37 °C. The fibers were dissolved through a gel-to-solution transition at room temperature (RT) below the LCST of 38 °C and thoroughly perfused with PBS wash. Each inlet and outlet was created by inserting luer tubes into the hydrogel. This setup facilitated media perfusion into the inlet, circulation through the microchannel network, and subsequent outflow through the outlet, creating a closed circulation system. The chip was cultured by perfusing each type of media at a flow rate of 4.71 mL min$^{-1}$ using a peristaltic pump (BT100-1L; Longer Precision

Pump Co., Ltd, Amersham, UK) with daily media change. Hypoxic media were generated by deoxygenating normoxic media through incubation in a Hypoxystation (Whitley H35; West Yorkshire, UK) with 0.1% $O_2$ and 5% $CO_2$ for 4 h. Microchannel visualization was achieved by perfusing green fluorescence microspheres (0.2 μm, TetraSpeck, T7280; Thermo Fisher Scientific) with PBS (1:250) under confocal imaging (LSM 980; Zeiss, Oberkochen, Germany) with Zen 3.3 blue edition. Hydrogel porosity was assessed using scanning electron microscopy (Merlin, Zeiss), followed by quantitative image analyses using ImageJ (Fiji).

## Computational fluidic dynamics (CFD)

CFD modeling was employed to determine the diffusion coefficient (D) of the hydrogel as a function of fiber density. It was also used to calculate the oxygen gradient within the chip based on cellular oxygen consumption. Initially, a single-inlet chip was used to experimentally derive the D in relation to the hydrogel porosity by varying the fiber density. This approach helped in isolating the effects of microchannel porosity from other parameters. To create an oxygen-free chip, the hydrogel was incubated in a Hypoxystation with 0.1% $O_2$ and 5% $CO_2$ for 4 h to remove oxygen. An oxygen dissolve meter (DM-1; CAS, Gyeonggi-do, Republic of Korea) was used to confirm the resulting anoxic conditions. The single-inlet chip was then perfused with normoxia media for 1 min while varying the fiber density (0, 1, 2, and 3 mg mL$^{-1}$). Oxygen profiling was performed at incremental distances from the inlet using a Microx 4 system (PM-PSt7; PreSens Precision Sensing GmbH, Regensburg, Germany). The 3D chip model was created using computer-aided design (CAD) software (SpaceClaim; ANSYS, Canonsburg, PA, USA). Two pairs (two-inlet chips) of cylindrical inlets and outlets, each 1 mm in diameter and 20 mm in length, were connected to a cuboid hydrogel with dimensions of 20 mm (width) × 20 mm (length) × 5 mm (thickness), consistent with the actual chip size. Finite element analysis via Fluent Meshing (2020R, ANSYS, USA) generated a total of 143,710 polyhedral mesh elements. The diffusion of oxygen gas within the chip hydrogel was calculated under continuous media (water) flow conditions using the media density (1007 kg m$^{-3}$). viscosity (0.958 × 10$^{-3}$ kg m$^{-1}$ s$^{-1}$), and the diffusion coefficient of oxygen-media (D$_{OM}$: 2.88 × 10$^{-9}$ m$^2$ s$^{-1}$ at 37 °C)[47]. This calculation assumed an equal diffusion coefficient for oxygen and the media. Darcy's law was used as the governing equation for hydrogel porosity, which is represented as:

$$\mathbf{Q} = -\boldsymbol{\kappa} \cdot A \cdot \Delta P / \mu \cdot L \qquad (1)$$

where **Q** denotes the flow rate, A signifies the cross-sectional area, L represents the length of the system, ΔP stands for the pressure drop, μ is the media viscosity, and **κ** represents the viscous resistance. According to Darcy's law, the pressure drop is proportional to the flow rate but inversely proportional to the viscous resistance of non-Newtonian flow, especially at low Reynolds numbers. The hydrogel's viscous resistance was set to 7.5 × 10$^{11}$ m$^{-2}$ [48], and porosity values of 38.8%, 44.4%, 54.6%, and 73% based on experimental data were used to reflect variations in fiber density (0, 1, 2, and 3 mg mL$^{-1}$), respectively. The inlet boundary conditions for CFD analysis included the inlet flow velocity (100 mm s$^{-1}$) and the oxygen concentration of normoxic media (5.45 mg mL$^{-1}$). Transient simulations of the single-inlet chip system were conducted over 3000 iterations with 0.1-s intervals. The resulting color-coded volume was rendered using chart visualization to illustrate the oxygen concentration gradient within the chip. Moreover, a regression curve was plotted to establish a correlation between oxygen levels (y-axis) and diffusion coefficients (x-axis). This facilitated the calculation of hydrogel diffusion coefficients as a function of the fiber density by inputting probe-recorded oxygen concentration values on the y-axis of the regression curve. The two-inlet chip with embedded HCC tissue was simulated by incorporating

tissue oxygen consumption into the CFD model. Tissue oxygen consumption was represented by a hypoxic oxygen influx modeled from the upper wall of the chip, allowing for the determination of the influx volume (Supplementary Fig. 3). The modeling was carried out using boundary conditions at the normoxia inlet ($O_2$ concentration = 5.45 mg mL$^{-1}$, inlet velocity = 100 mm s$^{-1}$), the hypoxia inlet ($O_2$ concentration = 0 mg mL$^{-1}$, inlet velocity = 0 mm s$^{-1}$), and the upper wall influx (0.007 mL s$^{-1}$). The calculation process went through 1000 iterations via steady-state simulation, with the results presented using color-coded contour diagrams and oxygen concentration plots. Additionally, the effects of oxygen diffusion on the viability of Hep3B cells were investigated by culturing 5 × 10$^6$ cells in the chip for 5 days. A CCK-8 assay (1:10 ratio, Dojindo Molecular Technologies, Inc., Rockville, MD, USA) was performed, and the absorbance readings were taken at 450 nm using a plate reader (SpectraMax Gemini™ XPS/EM; Molecular Devices LLC, San Jose, CA, USA).

## Tissue culture in the chip and immunohistochemistry

Tissue pieces (50 mg mL$^{-1}$) were obtained from HCC patient tissues using a biopsy punch (1 mm in diameter, BP-10F; Kai Medical, TX, USA). The pieces were mixed with gelatin hydrogel and embedded in the chip at 37 °C, followed by media perfusion for 4 days. Tissues were then fixed with 4% paraformaldehyde (PFA, CNP015-0550; CellNest, Hanam-si, Gyeonggi-do, Republic of Korea) in PBS for 2 h at RT, washed three times with PBS, and then embedded in paraffin before being sectioned into 4 μm thick blocks. Hematoxylin and eosin (H&E) staining was conducted according to standard protocols, and optical imaging was performed using an inverted microscope (DMi8 M; Leica, Wetzlar, Germany).

Deparaffinization and rehydration of the tissue sections were accomplished using a series of dilutions with xylene and ethanol (100, 95, 80, and 70% v v$^{-1}$ in deionized water). Antigen retrieval for CD31 was carried out using a low-pH buffer (k8005; Agilent Dako, Santa Clara, CA, USA), while a high-pH buffer (k800421-2; Agilent Dako) was used for K19, CD34, and AFP. CAIX did not require antigen retrieval. Endogenous peroxidases were deactivated by incubating the samples with a 3% $H_2O_2$ solution (H1009; Sigma-Aldrich) for 10 min, followed by washing with tris-buffered saline (TBS, ML023-03; Welgene, Gyeongsan-si, Gyeongsangbuk-do, Republic of Korea) and blocking with 5% bovine serum albumin (BSA, A0100-005; GenDEPOT, Altair, TX, USA) in PBS. The samples were then incubated with primary antibodies for CAIX (1:1000, NB100-417, Novus Biological LLC), CD31 (1:100, mouse monoclonal, NB600-562, Novus Biological LLC), K19 (1:1000, mouse monoclonal, ab9221, Abcam), CD34 (1:50, GA63261-2, Agilent Dako), and AFP (1:100, rabbit monoclonal, ab169552, Abcam) at RT for 1 h. Subsequently, the samples were incubated with HRP-labeled secondary antibodies (1:5000, anti-rabbit polymer, k4003; Agilent Dako) at RT for 20 min. Afterward, the samples were treated with DAB development solution (k3468; Agilent Dako) for 5 min, washed with deionized water, and counterstained with hematoxylin (k8008; Agilent Dako) before optical imaging. Cell apoptosis was assessed using the Click-iT Plus TUNEL assay (C10617; Thermo Fisher Scientific) following the manufacturer's instructions. Confocal imaging (LSM 980; Zeiss) was performed with Zen 3.3 blue edition, and quantitative image analysis was conducted using ImageJ (Fiji).

## Determination of cell preference to oxygen levels

HCC cells from typical (#6) and IRE (#9) HCC tissue were labeled with DiI (green) and DiO (red), respectively, using the Vybrant Multicolor cell-labeling kit (V22889; Thermo Fisher Scientific) at a ratio of 1:500 for 30 min. Subsequently, the cells were embedded and cultured in the chip at a concentration of 5 × 10$^6$ cells mL$^{-1}$ for 7 days. After harvesting the hydrogels, they were washed with PBS three times and subjected to confocal imaging using tile and z-stack scanning (LSM 980; Zeiss) with Zen 3.3 blue edition. Quantitative image analysis was conducted using

ImageJ (Fiji). Furthermore, HCC cells (typical #6 and IRE #9) were cultured in 24-well plates at a density of $5 \times 10^4$ cells well$^{-1}$ for 3 days under either normoxic (20% O$_2$) or hypoxic (0.1% O$_2$) conditions. Following culture, the cells were fixed with 4% PFA in PBS at RT for 30 min and washed three times with PBS. Then, the cells were permeabilized using 0.1% Triton X-100 (T8787; Sigma-Aldrich) in PBS at RT for 1 h, followed by another three washes with PBS. F-actin staining was achieved by treating the cells with Alexa Fluor 488 phalloidin (1:500, A12379; Thermo Fisher Scientific) at RT for 1 h, followed by three washes with PBS. Confocal imaging with Zen 3.3 blue edition and quantitative image analysis by ImageJ (Fiji) were performed after 4′,6-diamidino-2-phenylindole, dihydrochloride (DAPI, R37606; Thermo Fisher Scientific) counterstaining. HCC cells (IRE #8) were cultured in the chip for 7 days and fixed overnight at 4 °C with 4% PFA in PBS. After three washes with PBS, the cells were permeabilized for 10 min using 0.5% Triton X-100 under perfusion with a 26 G syringe, followed by incubation at RT for 1 h. Subsequently, the samples were blocked for 1 h at RT using 1% BSA and 0.2% Triton X-100 in PBS. Primary antibodies for CD34 (1:200, mouse monoclonal, sc-7324; Santa Cruz Biotechnology, Dallas, TX, USA) and CAIX (1: 500, rabbit monoclonal, NB100-417; Novus Biological LLC), serving as markers of microvascular invasion and stemness, respectively, were applied to the samples and incubated overnight at 4 °C. After three washes with PBS, the samples were treated with secondary antibodies: fluorescein (FITC)-conjugated AffiniPure (1:250, goat anti-mouse IgG, 115-095-003; Jackson Immuno Research, West Grove, PA, USA) for CD34 and Alexa Fluor 594-conjugated AffiniPure (1:250, goat anti-rabbit IgG, 111-585-003, Jackson Immuno Research) for CAIX for 2 h at RT. Following another three washes with PBS, the samples were stained with DAPI for nucleus visualization and subjected to confocal imaging (LSM 980; Zeiss) in z-stack and tile scanning with Zen 3.3 blue edition. Quantitative image analysis was performed using ImageJ (Fiji).

## qRT-PCR

Total RNA was extracted from cells using TRIzol reagent (15596018; Thermo Fisher Scientific), followed by reverse transcription of 1 μg of RNA into cDNA using AccuPower Cycle Script RT Pre-mix (Bioneer, Daejeon, Republic of Korea). The entire primer sequences were designed using the National Center for Biotechnology Information via Primer-BLAST. Oligonucleotides were purchased from Cosmogenetech (Seongdong-gu, Seoul, Republic of Korea). The StepOne Real-Time PCR system (Applied Biosystems, Foster City, CA, USA) was then employed for 40 cycles of target gene amplification with SYBR Green, cDNA, and primers (Table S2, Supporting Information). Gene expression levels were determined by comparing them with the $C_t$ value of the housekeeping gene glyceraldehyde-3-phosphate dehydrogenase (*GAPDH*) using StepOnePlus version 2.3. The resulting data were presented as a heatmap after converting the $C_t$ values into Log$_2$Fold values, and visualization was carried out using the RStudio (Version: 2023.06.1+524; Boston, MA, USA) heatmap.2 function.

## Western blot

Proteins were extracted through incubation of samples in RIPA lysis buffer (89900; Thermo Fisher Scientific) supplemented with 1X Protease and Phosphatase Inhibitor Cocktail (78440; Thermo Fisher Scientific) on ice for 2 h. This was followed by syringe shattering with needles of varying diameters (18–26 G). Protein concentrations were then determined using a BCA protein assay kit (23227; Thermo Fisher Scientific). The protein samples were loaded onto SDS-PAGE (4–15% w v$^{-1}$) polyacrylamide gel electrophoresis (4561094; Bio-Rad, Hercules, CA, USA) and then electro-transferred onto a nitrocellulose membrane using the iBlot 2 Gel Transfer Device (Invitrogen, Carlsbad, CA, USA). After blocking with a solution of 5% nonfat dry milk in TBS containing 0.1% Tween-20 (P9416; Sigma-Aldrich) for 1 h at room temperature, the membrane was incubated overnight at 4 °C with primary antibodies for CAIX (1:500, rabbit polyclonal, NB100-417; Novus Biologicals LLC, Centennial, CO, USA), ABCB1 (1:250, mouse monoclonal, sc-55510; Santa Cruz Biotechnology), and β-actin (1:500, mouse monoclonal, sc-47778; Santa Cruz Biotechnology). After three washes with TBS-T, the samples were exposed to secondary antibodies at a dilution of 1:5000. This included an anti-rabbit IgG antibody (ab6721; Abcam) for CAIX and an anti-mouse IgG antibody (ab6708; Abcam) for ABCB1 and β-actin. Protein bands on the membranes were observed using a chemiluminescence imaging system (ImageQuant LAS 4000; GE Healthcare Life Sciences, Chicago, IL, USA), and subjected to quantitative image analysis using ImageJ (Fiji).

## Proteinase (MMP1) activity

Proteins were initially extracted and stored at −80 °C, then homogenized using 100 μL of cell lysis butter. After a 5-min incubation on ice, the samples were subjected to centrifugation at $16,000 \times g$ at 4 °C for 10 min. The resulting supernatant was then transferred to a pre-chilled tube, and its concentration was determined using a BCA protein assay kit. To assess MMP1 activity, a fluorometric-based collagenase (collagen degradation/zymography) assay kit (ab234624, Abcam) was employed according to the manufacturer's instructions. Fluorescence intensity was measured using a fluorometer (Varioskan Flash 3001, Thermo Fisher Scientific) at an excitation/emission wavelength of 490/520 nm in the kinetic mode at 37 °C for 90 min with 4-min intervals.

## IC50 of anti-cancer drugs

HCC cells were cultured in 96-well plates at a density of $1 \times 10^4$ cells per well and allowed to reach 90% confluency over three days. Afterward, the cells were treated with Doxorubicin (D1515, Sigma-Aldrich) dissolved in distilled water at concentrations ranging from 0 to 1 μM or Sorafenib (SML2653, Sigma-Aldrich) dissolved in DMSO at concentrations ranging from 0 to 40 μM for 24 h at 37 °C with 5% CO$_2$. Cell viability was assessed using a CCK-8 assay, performed at a 1:10 dilution, and the IC50 value for each drug was determined using the trendline.

## Dual gradient chip

The dual gradient chip was developed by establishing a drug gradient perpendicular to the oxygen gradient. Initially, HCC tissues from IRE #8 and #9, at a concentration of 50 mg mL$^{-1}$, were cultured on the chip for 7 days under the oxygen gradient. Subsequently, the cells were exposed to drug perfusion through one inlet of the chip for 3 days, with the concentration set 3 times higher than their respective IC50 value. At the same time, drug-free media was perfused through the other inlet to establish the drug gradient. Cell viability was assessed using a live/dead assay (L3224; Thermo Fisher Scientific) followed by confocal imaging (LSM 980; Zeiss) with Zen 3.3 blue edition. The regions of interest (ROI) were analyzed using the ImageJ plugin, and visualization was performed using the RStudio heatmap.2 functions excluding the clustering step.

## Oxygen profiling in vivo

Male BALB/c nude mice aged six weeks, were obtained from Orient Bio (Seoul, Republic of Korea) and acclimated to a pathogen-free environment with a 12-h light/dark cycle, ambient temperature and humidity. Organ oxygen levels were assessed using the Microx 4 oxygen profiling system in three distinct groups: normal limb (NL), subcutaneous (SC), and ischemic limb (IL). Anesthesia was administered through intraperitoneal injection of zoletil (50 mg kg$^{-1}$) and xylazine (10 mg kg$^{-1}$). In the NL and SC groups, a small incision (<1 mm) was made in the right hindlimb and right subcutaneous region, respectively. In the IL group, ischemia was induced by ligating the femoral artery of the right hindlimb at the proximal and distal points using 4-0 silk, followed by resection of the artery segment between the ligation points. Subsequently, oxygen levels were measured using the system probe, and the mice were euthanized in a CO$_2$ chamber.

## Chip PDX model with systemic drug response

A microchannel hydrogel ($2 \times 2 \times 2$ mm) containing IRE #8 tissues (50 mg mL$^{-1}$), epithelial growth factor (EGF, 20 ng mL$^{-1}$), and basic fibroblast growth factor (b-FGF, 10 ng mL$^{-1}$), was surgically implanted into the NL, SC, and IL regions of mice for 6 weeks. Two weeks after the chip implantation, the mice were injected with Doxorubicin in distilled water (100 μL, 3 mg kg$^{-1}$) through the tail vein. This injection was repeated after 2 weeks, resulting in a total of two doses. Additionally, Sorafenib in DMSO (100 μL, 30 mg kg$^{-1}$) was orally administered to each mouse. Drug responsiveness was evaluated by comparing the effects of the vehicle control (consisting of no drugs in DMSO and saline) with each specific drug (Doxorubicin or Sorafenib). After euthanasia using $CO_2$ gas, tumor tissues were collected, and the tumor volume was measured using digital calipers. The x, y, and z-axis values were calculated and normalized to the volume of the tumor tissue at day 0, which served as the baseline (i.e., the volume of the mother tissue).

## Statistical analysis

Data analysis was conducted using Excel (version 16.0.17531.20004), SigmaPlot 12.0 (Systat Software Inc., San Jose, CA, USA) and Prism 8.0.1 (GraphPad Software, Boston, MA, USA). The data are presented as means ± standard error of the mean. Paired comparisons were assessed using a two-tailed Student's t-test, while multiple comparisons were carried out using one-way ANOVA with Tukey's significant difference post-hoc test. Statistical significance was indicated as exact $P$-value. Clinical data were evaluated using the t-test of independent samples, with equal variances pooled for typical and IRE patients. Events, such as viral infection and recurrence ratio, were analyzed using the chi-square test. Histological differences were examined using Fisher's exact test. Data were normalized and transformed with corresponding n numbers as described in each figure legend.

## Reporting summary

Further information on research design is available in the Nature Portfolio Reporting Summary linked to this article.

## Data availability

All figures and supplementary information data are available in the Figshare repository (https://doi.org/10.6084/m9.figshare.25585770.v1). The private clinical information is protected and is not available due to data privacy laws. Any additional requests for information can be directed to, and will be fulfilled by, the corresponding authors. Source data are provided with this paper.

## Code availability

The heatmap.2R code is accessible on GitHub via the following link: https://doi.org/10.5281/zenodo.11191336.

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

## Acknowledgements

This study was supported by the National Research Foundation of Korea(NRF) Grant funded by the Korean Government(MSIP) (No. NRF-RS-2023-00207857), awarded to HJS. Additionally, this study received support from a Faculty Research Grant from Yonsei University College of Medicine grant (6-2016-0122) and an NRF Grant (2021R1I1A1A01059932) awarded to H.D.H. The funding for the training component was provided by Brain Korea 21 PLUS Project for Medical Science at Yonsei University College of Medicine and Hur Jiyoung Foundation, awarded to S.W.B. The authors would like to express their gratitude for the support provided by Medical Illustration & Design, a unit within the Medical Research Support Services at Yonsei University College of Medicine, for enhancing the visual presentation of the content. The figure illustrations were created using BioRender.com.

## Author contributions

S.B. and H.H. conceptualized the study, devised the experimental methodology, performed the experimental studies, and formally analyzed the data. J.S.P. conducted the investigations. M.J.C. reviewed and edited the manuscript. H.S.K., S.E.Y., and S.C. provided resources for the research. C.K., J.K., J.Y.L., Y.L., H.K., Y.N., S.C., and K.L. performed the experimental studies. J.K.Y. and J.S.C. conceptualized the study. D.H.H. and H.J.S. drafted the original manuscript, supervised the project, and acquired funding for the project.

## Competing interests

The authors declare no competing interests.
