## [Peer Review File · Nature Communications]

REVIEWER COMMENTS

Reviewer #1 (Remarks to the Author):

This study establishes a novel culture system for hepatocellular carcinoma (HCC) that mimics oxygen gradients. The authors demonstrate that HCC cells in the low-oxygen region exhibit higher malignancy and invasiveness, contributing to increased recurrence. Furthermore, using this system, the authors analyze the drug sensitivity of HCC and reveal that HCC cells in the low-oxygen region are resistant to drugs. This study is highly original and innovative.

Minor comments:

1. Liver lobules are subdivided into 3 zonation, with the periportal zone called as zone 1 and the perivenous zone as zone 3. It is known that oxygen levels decrease from zone 1 to 3. Recent studies report that zone 3 hepatocytes have a higher carcinogenic potential (JHEP Rep. 2021 May 27;3(4):100315., Proc Natl Acad Sci U S A. 2019 Sep 24;116(39):19530-19540.). Cite these papers and discuss their relevance to the findings of this study.

2. Analysis using clinical specimens is limited by the small number of cases, preventing conclusive statements. Address this as a limitation of the current study.

Reviewer #2 (Remarks to the Author):

Dear authors,

The manuscript “Chip collection of hepatocellular carcinoma based on O₂ heterogeneity from patient tissue” describes a novel method for creating in vitro models of two different hepatocellular carcinoma types and validates in vitro findings from the described chip model against mouse models and patient outcomes.

Major comments:

1. The start of the introduction is very technical. It would be very useful, especially for the non-clinicians reading the paper, to introduce HCC in general and highlight some of the current clinical problems that the presented model can address.

2. Given that cells often change their identity, characteristics and function when cultured ex vivo, it would be useful to demonstrate increased amounts of quality control data for the here used human material (e.g., Cell type identity/function markers expressed over culture time/changes of those during culture), do the cells adjust to their culture environment over time or is the in vivo phenotype stable over extended periods ex vivo/in vitro?

3. Whilst several patient samples are used throughout the study it appears that only one patient of each HCC type was used for each experiment. It is difficult to draw conclusions from a lot of these experiments without adding additional patient samples per experiment to account for inter-patient variability. Please include.

4. Several observations are related to clinical observations in the follow-up of the patients from whom material was collected. These notions remain speculative and should be removed.

5. Given the data in Figure 3C showing significant loss of viability for the HCC types grown in hypoxic/normoxic conditions, an analysis of cell death in Figure 4 would be useful to demonstrate that the separation is due to migration and not due to loss of viability on the opposing sides.

6. Given this system is quite technically complex it would be useful to have controls with less variables to compare to, for example a chip grown purely in hypoxia, and one grown purely in normoxia for the IRE and normal HCC samples respectively to demonstrate the benefits of a more complex system.

7. In several sections the paper is difficult to follow and attention has to be given to the English language as well as a range of typos need to be corrected.

Minor comments:

1. The methods are a bit unclear for how the media recirculation and oxygenation works. Is the medium recirculated in a closed loop or from a supply reservoir to a waste reservoir? From my understanding, the deoxygenation appears to be performed prior to using the chips, which are then cultured for 24 hours between media changes. As the dissolved oxygen measurements are taken after 1 minute, are these data and the resulting CFD analysis relevant for the system over a 24-hour cycle, considering reoxygenation that will occur at a reasonable rate due to the convective flow? Please elaborate and include long-term measurements if needed.

2. Figure 2 iii and iv: Please include representable images for the here shown data (e.g., viability staining image example, representative images of a cell line at different stages, can be in supplementary data).

Response to Reviewers

We sincerely thank the editor and the reviewers for their thorough evaluation and consideration of our manuscript. Your time and expertise are greatly appreciated. The insightful feedback and constructive guidance provided have been invaluable in refining our work. We hope that the revised manuscript positively assists you and the reviewers in reaching a final decision. Enclosed in this document are detailed responses addressing each raised concern, along with the revised manuscript and Supplementary information. Our responses and the corresponding changes in the manuscript are highlighted in blue for clarity.

Reviewer #1 (Remarks to the Author):

[R1Q-Overall] This study establishes a novel culture system for hepatocellular carcinoma (HCC) that mimics oxygen gradients. The authors demonstrate that HCC cells in the low-oxygen region exhibit higher malignancy and invasiveness, contributing to increased recurrence. Furthermore, using this system, the authors analyze the drug sensitivity of HCC and reveal that HCC cells in the low-oxygen region are resistant to drugs. This study is highly original and innovative.

[R1A-Overall] The authors feel grateful for the positive comments from the reviewer. In response to these comments, the entire manuscript has been revised point-by-point with modification of each section as highlighted below in blue.

Minor comments:

[R1Q1] Liver lobules are subdivided into 3 zonation, with the periportal zone called as zone 1 and the perivenous zone as zone 3. It is known that oxygen levels decrease from zone 1 to 3. Recent studies report that zone 3 hepatocytes have a higher carcinogenic potential (JHEP Rep. 2021 May 27;3(4):100315., Proc Natl Acad Sci U S A. 2019 Sep 24;116(39):19530-19540.). Cite these papers and discuss their relevance to the findings of this study.

[R1A1] Thank you for the constructive comment. The authors have addressed the reviewer's comment by explaining the decrease in oxygen levels from zones 1 to 3 in liver lobules and by adding more references to the discussion section as follows.

[Discussion]

The liver is comprised of lobules as a form of hexagonal units, where three zones are divided

following the degree of oxygen supply. The periportal zone 1 surrounds the portal tracts and receives oxygenated blood from the hepatic arteries. On the other hand, the perivenous zone 3 is located adjacent to the central veins, where the oxygen supply is limited. Blood flows from the portal tract to the venules through the three zones, creating an oxygen gradient within the liver. Simulation of the hypoxic environment of perivenous zone 3 (Fig. 3) enables the detection and isolation of hepatic cells with carcinogenic potential^{1,2}, offering insights into HCC progression and therapeutic strategies. This heterogeneity significantly contributes to the low engraftment rates in the conventional culture and PDX models.

[References]

40. Kurosaki, S. et al. Cell fate analysis of zone 3 hepatocytes in liver injury and tumorigenesis. *JHEP Reports* 3, 100315 (2021).
41. Ang, C.H. et al. Lgr5+ pericentral hepatocytes are self-maintained in normal liver regeneration and susceptible to hepatocarcinogenesis. *Proceedings of the National Academy of Sciences* 116, 19530-19540 (2019).

[R1Q2] Analysis using clinical specimens is limited by the small number of cases, preventing conclusive statements. Address this as a limitation of the current study.

[R1A2] We absolutely agree to the comment. As a limitation of the study, a small number of cases was discussed through revision as follows.

[Discussion]

As the small number of HCC cases appeared to be a limitation of the present study, the next study plans to recruit a larger cohort of HCC patients to increase inter-patient heterogeneity and validate the chip approach across various cancer types. Furthermore, the incorporation of a large animal model is considered to enhance the reliability in simulating variations in both oxygen and drug concentrations so that more clinically meaningful data can be generated.

Reviewer #2 (Remarks to the Author):

[R2Q-Overall]

The manuscript “Chip collection of hepatocellular carcinoma based on O₂ heterogeneity from patient

tissue” describes a novel method for creating in vitro models of two different hepatocellular carcinoma types and validates in vitro findings from the described chip model against mouse models and patient outcomes.

[R2A-Overall] We appreciate the positive evaluation. The constructive suggestions instructed the authors to thoroughly revise the manuscript by addressing each point of concerns.

Major comments:

[R2Q1] The start of the introduction is very technical. It would be very useful, especially for the non-clinicians reading the paper, to introduce HCC in general and highlight some of the current clinical problems that the presented model can address.

[R2A1] We appreciate the suggestion and as a result, we have included more information on HCCs for non-clinicians in the revised introduction.

[Introduction]

Hepatocellular carcinoma (HCC) is the most common type of primary liver cancer and the second leading cause of cancer-related mortality worldwide, with its incidence on the rise^{3,4}. Although liver resection remains the preferred treatment for achieving long-term survival, recurrence poses a challenge for more than half of HCC patients^{5,6}. As a result, there is a growing interest in HCC avatar models to tailor clinical strategies specific to HCC types. Recent efforts have focused on establishing i) patient-derived xenograft (PDX) by implanting cancer cells into mice or ii) cell spheroid models as a form of organoid using patient cancer tissues⁷⁻⁹. However, these models often fail to actively replicate the diverse oxygen gradients found within the HCC tumor microenvironment.

[R2Q2] Given that cells often change their identity, characteristics and function when cultured ex vivo, it would be useful to demonstrate increased amounts of quality control data for the here used human material (e.g., Cell type identity/function markers expressed over culture time/changes of those during culture), do the cells adjust to their culture environment over time or is the in vivo phenotype stable over extended periods ex vivo/in vitro?

[R2A2] Thank you for the insightful comment. In our previous study, direct culture of patient-derived tissue in the chip system was demonstrated by maintaining the genetic characteristics of patient tissues over a 3-week period¹⁰. As this previous system lacked the oxygen gradient, the current

revision presented the result of gene profiling upon 10-day culture in the chip (Supplementary Fig. 10) with verification of the phenotypic maintenance. The result and discussion sections with the figure legend were revised accordingly as follows.

[Result]

On-chip separation and collection of HCC cells based on oxygen levels

When patient-derived cells were cultured in the chip for 10 days, the phenotypic gene profiles were maintained with respect to hypoxia (CAIX, VEGFR, GLUT1), drug resistance (ABCB1, ABCC2, GSTP1), ECM remodeling (CD44, Vim, MMP9), and HCC indicators (AFP) in the normoxic and hypoxic zones. Typical (#7) cells retained the genetic profile of mother tissue in the normoxic zone in contrast to the hypoxia-specific retention of IRE (#9) cells (Supplementary Fig. 10), indicating stable phenotypic maintenance of patient-derived cells upon chip culture for the experimental periods.

[Figure legend]

Supplementary Fig. 10 | 10-day maintenance of phenotypic gene profiles in patient cells upon chip culture. In the normoxic zone, typical (#7) cells retain the gene expression profile of mother tissue (0: white) with respect to hypoxia (CAIX, VEGF, GLUT1), drug resistance (ABCB1, ABCC2, GSTP1), and ECM remodeling (CD44, Vim, MMP9) and HCC indicator (AFP) for 10 days in the chip culture. In contrast, the hypoxia specific-maintenance (0: white) of mother tissue characteristic is dominantly seen in IRE (#9) cells. The results are presented as log₂fold change values with normalization to those of corresponding mother tissue.

[Discussion]

During 10 days in the chip culture (Supplementary Fig. 10), patient-derived cells underwent expression changes in the phenotypic genes that are associated with hypoxia, drug resistance, ECM remodeling, and HCC markers. Typical (#7) and IRE (#9) cells retained the mother tissue-like profiles in the normoxic and hypoxic zones, respectively, indicating the stable maintenance of cell characteristics by the chip for the experimental period. This claim was supported by the 10-day drug responses of patient tissues in the chip (Fig. 5) and the multi-spot chip PDX model for over 6-week period of implantation as the IRE and Typical characteristics were maintained when the histological features, gene profiling, and drug responses were analyzed (Fig. 6). Moreover, our previous study verified that direct culture of patient-derived tissue in the chip system with the absence of oxygen gradient resulted in stable maintenance of the genetic characteristics in patient tissues over a 3-week period¹⁰.

[R2Q3] Whilst several patient samples are used throughout the study it appears that only one patient of each HCC type was used for each experiment. It is difficult to draw conclusions from a lot of these experiments without adding additional patient samples per experiment to account for inter-patient variability. Please include.

[R2A3] We agree to the comment and have therefore made the following additions (Figure 4c and Supplementary Fig. 6,8,10 and 11) with the corresponding legends and result sections.

[Result]

Hypoxic IRE HCCs present challenges in the current HCC clinic

When HCC patient tissues were cultured on the oxygen gradient chip for 4 days (Fig. 3b, Supplementary Fig. 6), typical HCC showed increased expression of protein markers associated with hypoxia-mediated stemness (CAIX), cancer invasiveness (K19), and microvascular invasion (CD34) on the normoxic side compared to the hypoxic side. Conversely, IRE HCC exhibited increased marker expression in the hypoxic zone relative to the normoxic zone, confirming that the chip effectively preserved the inherent tissue characteristics related to oxygen preference.

On-chip separation and collection of HCC cells based on oxygen levels

For 7 days, the HCC cells were simultaneously perfused with normoxic (20% O₂) and hypoxic (0.1% O₂) media through the two inlets. Confocal imaging revealed that the IRE cells (red DiI) gradually migrated to the hypoxic side, while the typical cells (green DiO) remained on the normoxic side. The chip effectively separated and collected the two types of HCC cells on their respective sides, as indicated by the day-7 histogram (Supplementary Fig. 8a).

On day 7 (Fig. 4c), marker gene expression related to hypoxia (CAIX, VEGF, GLUT1) and drug resistance (ABCB1, ABCC2, GSTP1) increased on the hypoxic side of the chip compared to day 4, while the expression levels on the normoxic side generally decreased. The differential expression patterns became more uniform on both sides, indicating the progressive separation of HCC types (Typical: #4, #5, #6 and IRE: #7, #8, #9) in preparation for collection.

When patient-derived cells were cultured in the chip for 10 days, the phenotypic gene profiles were maintained with respect to hypoxia (CAIX, VEGFR, GLUT1), drug resistance (ABCB1, ABCC2, GSTP1), ECM remodeling (CD44, Vim, MMP9), and HCC indicators (AFP) in the normoxic and

hypoxic zones. Typical (#7) cells retained the genetic profile of mother tissue in the normoxic zone in contrast to the hypoxia-specific retention of IRE (#9) cells (Supplementary Fig. 10), indicating stable phenotypic maintenance of patient-derived cells upon chip culture for the experimental periods.

Dual gradient chip for hypoxia-targeted collection of drug-resistant IRE cells

After a 7-day culture of HCC patient tissues (#3, #8, #9) using a two-inlet perfusion system, the oxygen gradient chip was able to separate drug-resistant IRE cells to the hypoxic side. The protein marker expression of microvascular invasion (CD34) and stemness (CAIX) across the chip provided evidence of the separation (Fig. 5a, Supplementary Fig. 11).

[Figure legend]

Supplementary Fig. 6 | Discerning HCC type-dependent preference of oxygen level by the chip.

When HCC patient tissues are cultured on the chip for 4 days, typical (#7) and IRE (#3) HCCs show the increased expression of protein markers associated with hypoxia-mediated stemness (CAIX), cancer invasiveness (K19), and microvascular invasion (CD34) on the normoxic and hypoxic sides, respectively, compared to the corresponding mother tissue and the opposite side. The results confirmed that the chip effectively preserves the inherent tissue characteristics related to oxygen preference (Scale bar = 200 μ m).

Supplementary Fig. 8 | On-chip separation and collection of viable HCC cells based on oxygen levels.

a. A mixture of typical (T) and IRE (I) HCC cells (Top: T#4+I#7 or Bottom: T#5+I#8) is cultured on the chip for 7 days. Confocal imaging shows that over time, IRE cells (red DiI) progressively migrate to the hypoxic side (to the right), while the typical cells (green DiO) remain predominantly on the normoxic side (to the left). After the 7-day culture period, the two types of HCC cells are separated and collected on the normoxic and hypoxic sides, respectively, as shown in the day-7 histogram (n=3, Scale bar = 1 mm). **b.** During the 7-day collection period in the chip, the HCC cells remain viable without experiencing significant cell death (Scale bar = 100 μ m).

Fig. 4. On-chip separation and collection of HCC cells based on O₂ levels. **c.** On day 7, marker gene expression related to hypoxia (CAIX, VEGF, GLUT1) and drug resistance (ABCB1, ABCC2, GSTP1) increases on the hypoxic side of the chip compared to day 4, while the expression levels on the normoxic side generally decrease. The differential expression patterns become more uniform on both sides, indicating the progressive separation of HCC types (Typical: #4, #5, #6 and IRE: #7, #8, #9) in preparation for collection. The results are presented as log₂fold change values with normalization to day 0, which are visualized using the heatmap function in RStudio (n=3).

Supplementary Fig. 10 | 10-day maintenance of phenotypic gene profiles in patient cells upon chip culture. In the normoxic zone, typical (#7) cells retain the gene expression profile of mother tissue (0: white) with respect to hypoxia (CAIX, VEGF, GLUT1), drug resistance (ABCB1, ABCC2, GSTP1), and ECM remodeling (CD44, Vim, MMP9) and HCC indicator (AFP) for 10 days in the chip culture. In contrast, the hypoxia specific-maintenance (0: white) of mother tissue characteristic is dominantly seen in IRE (#9) cells. The results are presented as log₂fold change values with normalization to those of corresponding mother tissue.

Supplementary Fig. 11 | Retrieval of IRE cells from HCC tissues using the chip. **a.** After 7-day culture of HCC patient tissue (#3 or #9) under dual inlet perfusion, the oxygen gradient chip enables collection of IRE cells in the hypoxic region with the incremental marker expression of microvascular invasion (CD34) and stemness (CAIX) (Scale bar = 1 mm) **b.** as confirmed by magnification (Scale bars = 100 μm).

[R2Q4] Several observations are related to clinical observations in the follow-up of the patients from whom material was collected. These notions remain speculative and should be removed.

[R2A4] Following the insightful feedback Figures 3d and 3e were removed along with their related results, methods, and discussion sections through revision. We discussed a limitation regarding the clinical sample size by elaboration future studies as follows.

[Discussion]

As the small number of HCC cases appeared to be a limitation of the present study, the next study plans to recruit a larger cohort of HCC patients to increase inter-patient heterogeneity and validate the chip approach across various cancer types. Furthermore, the incorporation of a large animal model is considered to enhance the reliability in simulating variations in both oxygen and drug concentrations so that more clinically meaningful data can be generated.

[R2Q5] Given the data in Figure 3C showing significant loss of viability for the HCC types grown in hypoxic/normoxic conditions, an analysis of cell death in Figure 4 would be useful to demonstrate that the separation is due to migration and not due to loss of viability on the opposing sides.

[R2A5] We appreciate the constructive comment and thus have conducted live/dead assays (Supplementary Fig. 8b) to explain the cell death in the tissue (Figure 3c) in conjunction with the cell migration and proliferation (Figure 4a-b).

The cell death % (Fig. 3c) was driven mainly because HCC cells stayed in tissues and were not able to migrate for survival under the preferred oxygen level. Thus, the y-axis of Fig. 3c was revised to “Death (%) of remained cells in tissue” accordingly with revision of the corresponding legend part (see below).

The result section was revised with addition of Supplementary Figure 8b and legend as follows.

[Results]

On-chip separation and collection of HCC cells based on oxygen levels

The cell death was promoted when the cells remained in the tissues without migration to the preferred oxygen condition (Fig. 3c). However, when a mixture of typical and IRE (#4+#7 or #5+#8) was cultured without tissues as a free migration condition (Supplementary Fig. 8b), the cell viability did not decrease from day 0 to 4 and further to 7 upon live and dead assays. The results indicate the cell migration to the HCC type-specific oxygen condition for survival (Fig. 4a) with the consequent high rate of proliferation (Fig. 4b) was not affected by the death rate when the cell migration was not hindered by remaining in the tissue (Fig. 3c).

[Figure legend]

Fig. 3. Hypoxic IRE HCCs as a challenge in the current HCC clinic. **c.** The patient tissues are cultured in the oxygen gradient chip for four days and then subjected to a TUNEL assay to analyze the death (%) of remained cells in the sample tissue. Consistent with the MRI findings, typical HCC shows higher variability on the normoxic side, while IRE HCC (n=3) exhibits preserved viability under hypoxic conditions.

Supplementary Fig. 8 | On-chip separation and collection of viable HCC cells based on oxygen levels. **a.** A mixture of typical (T) and IRE (I) HCC cells (Top: T#4+I#7 or Bottom: T#5+I#8) is cultured on the chip for 7 days. Confocal imaging shows that over time, IRE cells (red DiI) progressively migrate to the hypoxic side (to the right), while the typical cells (green DiO) remain predominantly on the normoxic side (to the left). After the 7-day culture period, the two types of HCC cells are separated and collected on the normoxic and hypoxic sides, respectively, as shown in the day-7 histogram (n=3, Scale bar = 1 mm). **b.** During the 7-day collection period in the chip, the HCC cells remain viable without experiencing significant cell death (Scale bar = 100 μ m).

[R2Q6] Given this system is quite technically complex it would be useful to have controls with less variables to compare to, for example, a chip grown purely in hypoxia, and one grown purely in normoxia for the IRE and normal HCC samples respectively to demonstrate the benefits of a more complex system.

[R2A6] Following the insightful suggestion, a mixture of typical HCC (patient #4) and IRE HCC (patient #7) cells was cultured under either uniform hypoxia or uniform normoxia for 7 days (Supplementary Fig. 9). The result was described with the legend as follows

Interestingly, our observations revealed no migration through the microchannel structure. Also, as detailed in Supplementary Fig. 10, typical HCC cells showed an increase in the expression of genes associated with hypoxia, drug resistance, and ECM remodeling under hypoxic conditions, while IRE HCC cells displayed decreased expression under normoxic conditions. The challenge with the conventional uniform culture system is its inability to separate cells that contribute to recurrence. Additionally, changes in gene expression under these conditions lead to the loss of cells' original tissue characteristics, further complicating the characterization process. These findings are presented in Supplementary Fig. 9 and have been incorporated into the revised results section for clarity.

[Results]

On-chip separation and collection of HCC cells based on O₂ levels

In contrast, no clear separation of cells was resulted under uniform normoxia or uniform hypoxia without the gradient, indicating a key role of oxygen gradient in separation of HCC type-specific cells into their preferred oxygen condition (Supplementary Fig. 9).

[Figure legend]

Supplementary Fig. 9 | No separation of HCC cells by uniform oxygen environments. A mixture of typical HCC cells (#4, green) and IRE HCC cells (#7, red) is cultured on the under uniform normoxia or uniform hypoxia for 7 days, resulting in no clear separation under confocal imaging. The result is validated by the histograms for days 0, 4, and 7. (n=3, Scale bar = 1mm).

[R2Q7] In several sections the paper is difficult to follow and attention has to be given to the English language as well as a range of typos need to be corrected.

[R2A7] The revised manuscript was carefully read multiple times, spell-checked, and professionally edited (blue) for the presentation by a new co-author whose first language.

Minor comments:

[R2Q8] The methods are a bit unclear for how the media recirculation and oxygenation works. Is the medium recirculated in a closed loop or from a supply reservoir to a waste reservoir? From my understanding, the deoxygenation appears to be performed prior to using the chips, which are then cultured for 24 hours between media changes. As the dissolved oxygen measurements are taken after 1 minute, are these data and the resulting CFD analysis relevant for the system over a 24-hour cycle, considering reoxygenation that will occur at a reasonable rate due to the convective flow? Please elaborate and include long-term measurements if needed.

[R2A8] We have revised the following sections to clearly explain the closed loop system and operation mechanism.

[Online Methods]

Production of the oxygen gradient chip

Each inlet and outlet was created by inserting luer tubes into the hydrogel. This setup facilitated media perfusion into the inlet, circulation through the microchannel network, and subsequent outflow through the outlet, creating a closed circulation system.

[Results]

Two-inlet CFD modeling with experimental validation

The oxygen levels in both the normoxic and hypoxic media remained stable during the perfusion culture (Supplementary Fig. 4). In particular, the hypoxic medium consistently maintained the oxygen concentration below 0.35 mg L^{-1} before media change, ensuring the hypoxic environment. This hypoxic oxygen level is lower than the physiological range in the portal vein ($0.6\text{--}0.7 \text{ mg L}^{-1}$) although there was a gradual increase in the overall trend of oxygen level, suggesting the need of media exchange every 24 hours.

[Figure legend]

Supplementary Fig. 4 | Monitoring oxygen level changes in the medium during perfusion culture. Dissolved oxygen levels are monitored continuously for 24 hours in the normoxic and hypoxic media under perfusion into the oxygen gradient chip. Minimal fluctuations are detected in the normoxic media. The hypoxic medium consistently maintains the oxygen concentration below

0.35 mg L⁻¹, which is lower than the physiological range in the portal vein (0.6~0.7 mg L⁻¹). However, there is a gradual increase in the overall trend of oxygen level, suggesting the need of media exchange every 24 hours.

[R2Q9] Figure 2 iii and iv: Please include representable images for the here shown data (e.g., viability staining image example, representative images of a cell line at different stages, can be in supplementary data).

[R2A9] We agree to the comment and thus have made the following revisions of result section and legend (Supplementary Fig. 5).

[Results]

Two-inlet CFD modeling with experimental validation

The viability of Hep3B cells significantly increased with increasing fiber density, as assessed by the CCK-8 assay and Live/Dead staining (Fig. 2j, Supplementary Fig. 5). Notably, the group without fibers exhibited significant cell death, highlighting the effectiveness of the chosen fiber density of 3 mg mL⁻¹.

[Figure legend]

Supplementary Fig. 5 | Improvement of cell viability by increasing the fiber density. The viability of Hep3B cells is improved upon live and dead assays as the fiber density increases during the 5-day perfusion culture (0 to 3 mg mL⁻¹, n=3, Scale bars = 200 μm). Data = mean ± standard deviation ***P* < 0.01 and ****P* < 0.001 versus between lined groups.

Reference

1. Kurosaki, S. et al. Cell fate analysis of zone 3 hepatocytes in liver injury and tumorigenesis. *JHEP Reports* **3**, 100315 (2021).
2. Ang, C.H. et al. Lgr5+ pericentral hepatocytes are self-maintained in normal liver regeneration and susceptible to hepatocarcinogenesis. *Proceedings of the National Academy of Sciences* **116**, 19530-19540 (2019).
3. Hepatocellular carcinoma. *Nature Reviews Disease Primers* **7**, 7 (2021).
4. Llovet, J.M. et al. Immunotherapies for hepatocellular carcinoma. *Nature Reviews Clinical Oncology* **19**, 151-172 (2022).
5. Tsilimigras, D.I. et al. Recurrence patterns and outcomes after resection of hepatocellular carcinoma within and beyond the Barcelona clinic liver cancer criteria. *Annals of surgical*

- oncology* **27**, 2321-2331 (2020).
6. Marrero, J.A. et al. Diagnosis, staging, and management of hepatocellular carcinoma: 2018 practice guidance by the American Association for the Study of Liver Diseases. *Hepatology* **68**, 723-750 (2018).
 7. Broutier, L. et al. Human primary liver cancer–derived organoid cultures for disease modeling and drug screening. *Nature medicine* **23**, 1424-1435 (2017).
 8. Portillo-Lara, R. & Annabi, N. Microengineered cancer-on-a-chip platforms to study the metastatic microenvironment. *Lab on a chip* **16**, 4063-4081 (2016).
 9. Wilson, G.K., Tennant, D.A. & McKeating, J.A. Hypoxia inducible factors in liver disease and hepatocellular carcinoma: current understanding and future directions. *Journal of hepatology* **61**, 1397-1406 (2014).
 10. Yoon, S.J. et al. Tissue Niche Miniature of Glioblastoma Patient Treated with Nano-Awakeners to Induce Suicide of Cancer Stem Cells. *Advanced Healthcare Materials* **11**, 2201586 (2022).

REVIEWERS' COMMENTS

Reviewer #2 (Remarks to the Author):

Thank you for so thoroughly addressing the concerns raised. I find the paper significantly improved.